# Bradoriids (Arthropoda) and the Cambrian Diversification

Mark A. S. McMenamin 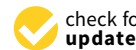

Department of Geology and Geography, Mount Holyoke College, South Hadley, MA 01075, USA;
mmcmenam@mtholyoke.edu; Tel.: +413-538-2280

**Abstract:** Bradoriids, among the earliest arthropods to appear in the fossil record, are extinct, ostracod-like bivalved forms that ranged from the early Cambrian to the Middle Ordovician. Bradoriids are notable for having appeared in the Cambrian fossil record before the earliest trilobites, and considering their rapid ascent to high genus-level diversity, provide key data for our understanding of the evolutionary dynamics of the Cambrian Explosion. This paper presents a broad review of bradoriid paleobiology. It is hypothesized here that an allele of *Antennapedia* determines whether bradoriid shields are preplete, amplete, or postplete. The preplete configuration of the shields of *Cambroarchilocus tigris* gen. nov. sp. nov. suggests that shield rowing motion may have propelled the animal backwards. Arcuate scars attributed here to a microdurophagous predator (*Arcuoichnus pierci* nov. ichnogen. nov. ichnosp.) occur on the paratype of *Cambroarchilocus tigris* gen. nov. sp. nov.

**Keywords:** Arthropoda; Bradoriida; Cambrian; Chengjiang; Sonora; México; *Antennapedia* complex; durophagous predation; ichnofossils

## 1. Introduction

Bradoriids (Bradoriida Raymond, 1935 [1]) are small bivalved basal euarthropods that appeared in shallow marine habitats during Cambrian Stage 3. The largest known bradoriid, *Petrianna*, is 17 mm long [2]. Bradoriids show a variety of modes of preservation (phosphatization being the most common [3–8]), and are particularly abundant in the Chengjiang Lagerstätte. Bradoriids such as *Sinskolutella* and *Yakutingella* are thought to have been detritophagous feeders [8] or possibly meiofaunal micropredators [5].

Bradoriids enter the fossil record before the first trilobites (shortly before the appearance of trilobites of the *Abadiella* Biozone) [3–5], and as such provide important data regarding the dynamics of the Cambrian Explosion. They are among the earliest arthropods recognizable in the fossil record [6]. Plausible inferences suggest that bradoriids may have an even earlier (pre-Atdabanian or Cambrian Stage 2) history [6], with Williams and coauthors [5] assuming that bradoriids had undergone significant evolution before their near simultaneous appearance in different parts of the world.

This pattern of early appearance and rapid diversification in bradoriids has been remarked on since Raymond C. Moore's 1961 diagram entitled "Stratigraphic distribution of archaeocopid ostracodes (*sic*)" [7], which shows 75% of the dozen listed genera as appearing at the base of the Cambrian. Paleontological interest has been building for the group, showing an accelerating increase in described genera in recent decades (Figure 1). The raw data for this graph are the description dates in the section of this article on bradoriid genera, as seen below.

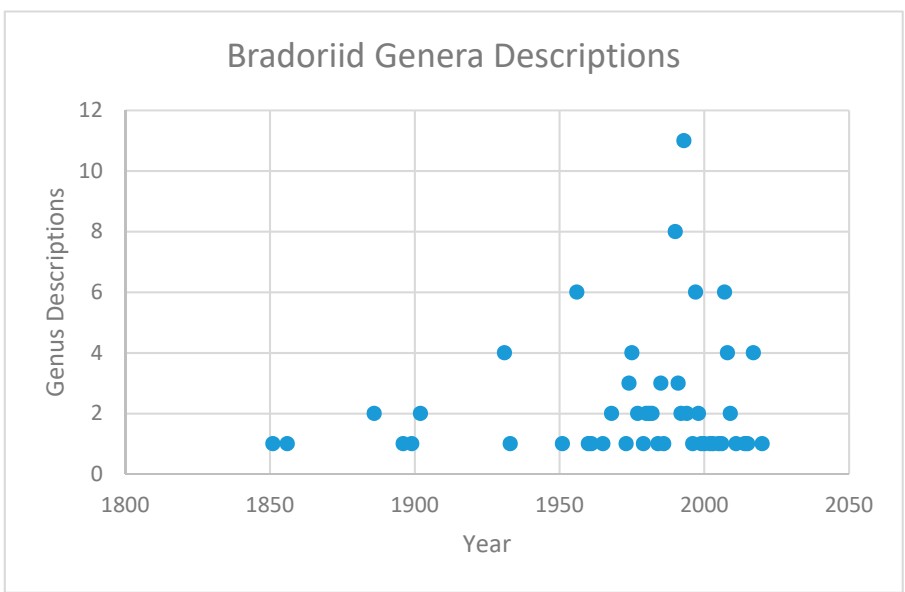

**Figure 1.** Descriptions of new bradoriid genera by year. Data from "Bradoriid Genera" section below.

The structure of the data in Figure 1, even after accounting for differences over time in the way bradoriids are studied, is interesting, showing a distinction between "background" descriptions and "mass descriptions", the latter reflecting primarily the publication of major monographic studies. Mass descriptions have generated increasing peaks in new genus descriptions, with a recurrence interval of roughly 25–35 years. Background descriptions remained steady for a century then began to climb steadily beginning in the 1960s. Neither background nor mass descriptions appear to have reached a plateau, suggesting that many more bradoriids remain to be discovered.

Bradoriids show an interesting pattern of diversification and extinction that ends in the Middle Ordovician [2]. Along with trilobites, they constitute part of a dual rapid diversification that is an important element of the Cambrian Explosion [5]. A spindle diagram of bradoriid diversity at the genus level (Figure 2; data from [2] and the section below on bradoriid genera) shows an explosion of diversity in Cambrian Stage 3, followed by a general decline in diversity (with the exception of a late diversification in late Stage 5 or Wuliuan) until the Early Ordovician.

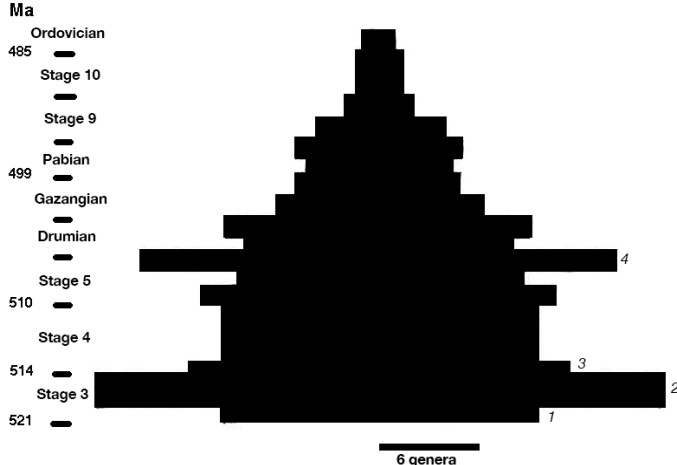

**Figure 2.** Genus-level bradoriid diversity during the Cambrian and Ordovician. Geological dates in million years (**Ma**). Stages refer to stages of the Cambrian. Numbers to the right indicate diversification waves 1–4. Data from [2] and "Bradoriid Genera" section below.

Fossil bivalved arthropods in general "provide a powerful tool to investigate the origins and dynamics" of evolutionary diversification [9]. A closer look at the bradoriid diversification in Cambrian Stage 3 reveals some interesting evolutionary dynamics. If we read the bradoriid fossil record literally in Stage 3, we see three waves of diversification during the Cambrian Arthropod Radiation Isotopic Excursion (CARE [2]). Each of these radiation bursts (named here Waves 1–3, with the late Wuliuan radiation as Wave 4) consists of a mix of relatively short-lived genera and long-lived genera (Table 1). Short-lived genera are largely confined to a single Cambrian stage, whereas long-lived taxa have ranges that extend significantly beyond a single stage (see Figure 4 in Williams et al. [2]).

**Table 1.** Bradoriid Radiations. Data from [2].

|  | **Short-lived Taxa** | **Long-lived Taxa** |
|---|---|---|
| Wave 1 (base of Stage 3) | 11 (55%) | 9 (45%) |
| Wave 2 (middle Stage 3) | 16 (76%) | 5 (24%) |
| Wave 3 (late Stage 3) | 11 (73%) | 4 (27%) |
| Wave 4 (late Wuliuan) | 12 (92%) | 1 (8%) |

These successive episodes of diversification, here called "waves", are interpreted as actual evolutionary events rather than artifacts of monographic swelling, as stratigraphic control [2] seems sufficient to allow recognition of closely spaced diversification events. An alternate view, namely, that all of these ranges actually extended to the same temporal/stratigraphic horizon [6] remains plausible and should not be dismissed out of hand in spite of the problems (in terms of phylogenetic telescoping, that is, originations of putative ancestor and descendant taxa piling up on the same stratigraphic horizon) that it poses for understanding the Cambrian evolutionary event [10].

The proportion of long-lived taxa originating during successive "waves" shows a general (if not monotonic) decrease from Cambrian Stage 3 to Wuliuan, dropping from an initial 45% to 8%. This is interesting because it would seem to be the inverse to the usual evolutionary expectation, namely, that later diversifications should generate longer-lived taxa, as more time had elapsed to allow improvements to the general "competitiveness" of the taxa involved.

To explain the interesting pattern noted in the previous paragraph, one might argue that, with the Cambrian predator revolution [11] underway, it was becoming more difficult for new bradoriid lineages to survive over the long term. This does not explain, however, why Wave 1 generated such a proportionately large number of long-lived taxa. The founder effect may have some explanatory power here; however, a putative founder effect does not explain why some of the very first bradoriid genera remained competitive over the (relative) long term in the face of three additional waves of new taxa. Proffering an explanation based on 'environmentally-cued polymorphism' [9] would not explain how Waves 1–3 came to be so closely spaced, seemingly without sufficient time to allow successive rapid diversifications.

Bradoriids apparently went extinct sometime during the Ordovician [12]. Some representatives may have persisted until the end-Ordovician extinction, although the latest records are Middle Ordovician and include a Siberian form (with the marvelously alliterative binomial) *Chegetella chegitunica* [13].

## 2. Bradoriid Diversification

Traditionally and conventionally classified as early ostracods [7], bradoriids lack "specialised post-antennal head appendages and are therefore not Eucrustacea" [2]. The affinities question has been ongoing for some time, with Zittel [14] noting that the Bradoriidae had once been classified as Cambrian Ostracoda but were subsequently removed to other groups such as Branchiopoda [14].

The present-day alternatives are to classify them as stem-group Crustacea [15] or basal/stem euarthropods [16,17]. This revised placement is based on relatively "few strong characters" and it is not certain that bradoriids belong within crown Mandibulata [18]. Edgecombe and Legg note that the "misleading resemblance of the bivalved carapace to that of ostracods" is also shared by other bivalved arthropods from the Cambrian [18].

Phosphatocopids, a Cambrian group like bradoriids also once referred to the Ostracoda, are distinguished from bradoriids by "the presence of a doublure on the free margin" [6]. Exceptions to this may occur, however, as the Australian bradoriid *Zepaera jagoi* shows a clear doublure/duplicature at the edge of the shield [19]. The majority of bradoriids have postplete or amplete shield shape, although a few are preplete (Figure 3); preplete shape is more characteristic for phospatocopids [20,21].

The list of Cambrian bivalved arthropods is extensive. A partial listing includes bradoriids, phosphatocopids, *Tuzoia*, *Branchiocaris*, *Isoxys* (once placed with question in the Bradoriidae [14]), *Yunnanocaris*, *Occacaris*, *Forfexicaris*, *Waptia*, *Clypecaris*, *Combinivalvula*, *Odaraia*, *Pectocaris*, *Mengdongella*, and *Jugatocaris*. In fact, the bivalved state may be a plesiomorphic character for Arthropoda [5]. Many of these groups were originally assigned to Phyllocarida or Branchiopoda based on valve morphology, but these comparisons did not withstand scrutiny once limb morphology became better known [18]. The subsequent removal of some of these groups from the crustacean crown- or even the euarthropod crown-group [18] recalls previous proposals regarding arthropod polyphyly [22]. Arthropods are considered here to be monophyletic [17], but their astonishingly rapid class-level diversification at the base of the Cambrian adds to the perceived and puzzling magnitude of the Cambrian Explosion.

Although arguments have been presented that bradoriids are polyphyletic [5,23], Bradoriida is treated here as monophyletic [6]. If this is indeed the case, and with additional information regarding the limb structure of bradoriids, erection of a new arthropodan class to accommodate this clade may be warranted.

Williams and coauthors [5] argued that the bradoriid families Beyrichonidae, Bradoriidae, Cambriidae, Comptalutidae, Hipponicharionidae, Kunmingellidae, and Svealutidae were united by three features: 1, a bivalved shell; 2, an entire, continuous latero-admarginal ridge or rim, and 3, postplete or amplete (but not proplete) shields (Figure 3). In addition, cambriids and kunmingellids have comparable lobation patterns and an anterodorsal cusp [5]. The lobes in cambriids and kunmingellids may be connected by ridges. Bradoriidae, Comptalutidae, and Svealutidae all tend to have a concentration of lobes in the anterodoral region [5]. Beyrichonidae, Cambriidae, Hipponicharionidae, and Kunmingellidae may be close relatives because of frequently conjoined, well delimited anterior and posterior lobes [5].

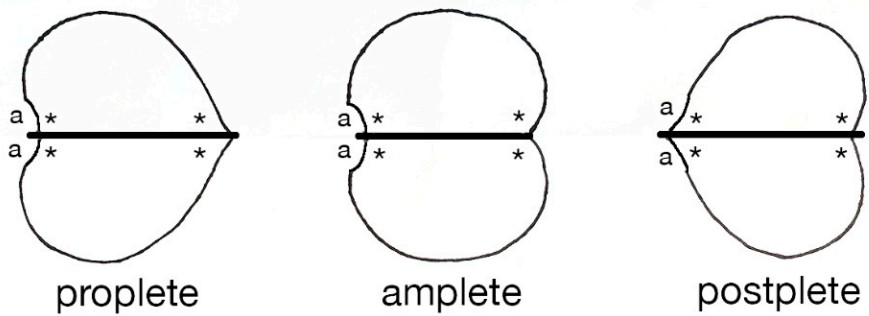

**Figure 3.** Proplete, amplete, and postplete bradoriid shield pairs. Most bradoriids develop postplete shields. Anterior to the left. Asterisk (*) denotes cardinal corners; "a" denotes anterodorsal cardinal corner marginal concavity. The relative height of each shield is measured by the distance it extends from the hinge line (black horizontal line); all three of the examples shown here have the same shield height. Image Credit: Mark McMenamin.

Although sexual dimorphism can be problematic to identify in the fossil record, it is thought to occur in bradoriids, with females in some species having a greater carapace height in relation to carapace width [24]. Bradoriids may have had as many as nine molt stages, and their assemblages tend to be monotypic [24]. Bradoriid species are also known for their high degree of morphological variability [19].

Interpretation of bradoriids is complicated by two factors. First, generally speaking only the bivalved carapace is known for any particular species; preserved appendages are rarely seen. Most species are known only from carapace morphology; classifying bradoriids (or ostracods, for that matter) on valve morphology alone poses special challenges [18]. These challenges can be addressed in some cases by careful attention to ontogeny [25]. It has been said that carapace "morphology is a poor means of establishing the affinity of bivalved arthropods, because the bivalved shield is a feature prone to evolutionary convergence" [16]. Furthermore, the reasons for changes in nodosity (i.e., the presence of external lobes and nodes on the valves) are not well understood and the same applies to ostracods [9].

## 3. Cephalic Appendages in Bradoriids

Bradoriids were once considered to be ancestral to modern ostracods, or classified as ostracods themselves, but the discovery of specimens with soft-bodied limb preservation at Chengjiang demonstrated that they are in fact an arthropod stem group [3,16]. Bradoriids never underwent the extreme desegmentation of the Ostracoda [3,22]. Desegmentation became nearly total in the podocopid ostracods [9]. In terms of appendage preservation and chaetotaxy, *Kunmingella douvillei* [3,16] is the best-known bradoriid. From anterior to posterior, the appendages consist of uniramous antennae (1), biramous appendages (2–8), and uniramous appendages (9–10) [18]. Five pairs of cephalic appendages (a1–a5) were identified [16] in *Kunmingella douvillei*. Although the pectinate anterior appendage structures in *Kunmingella douvillei* (= *K. maotianshanensis*) were identified as exopodite cephalic appendages (*ex.c app*), it seems likely that they functioned in a way similar to structures in ostracods that are of endopodite origin. For example, the modern ostracod *Cypria petenensis* has comparable pectinate structures projecting from its mandibular endopodite [26].

## 4. Bradoriid Locomotion

The bradoriid hinge has no sockets nor teeth as do the beyrichiocopid and podocopid ostracods of the Ordovician [5]. The ends of the hinge at the anterior of each shield are called the cardinal corners (Figure 3). Recognizing the anterior versus posterior orientation of bradoriid shields can be challenging [7], and this problem is longstanding. Charles D. Walcott remarked in 1887 [27] that it "is difficult to determine the anterior and posterior ends of the valves in many of the specimens." Sometimes a saw-like row of teeth marks the posterior margin. Recognition of what is here called the *anterodorsal cardinal corner marginal concavity* helps to distinguish the anterior ends of shields, and works consistently for preplete, amplete and postplete taxa (Figure 3). This feature is associated with the anterior cardinal angle as shown in Siveter and Williams [20], and is well displayed in *Bradoria scrutator* [20]. An excellent example of the anterodorsal cardinal corner marginal concavity is seen in preplete *Dielymella recticardinalis* where the anterior cardinal corner is as well drawn out into a short spine [20]—the anterodorsal cardinal corner marginal concavity clearly indicates the anterior end of the bradoriid. In cases where there is a prominent posterodorsal cardinal corner concavity as in *Mongolitubulus athelios*, the anterior-posterior orientation is often indicated by both the sweep of the great spines projecting from the mid-region of each shield, and also by the short spines along the posteroventral margin.

The position of the hinged bradoriid carapace in life has been controversial, with some advocating that the valves were held open in a "butterfly position", with others [28] arguing that the valves could be tightly sealed with all appendages protected inside. In *Cambroarchilocus tigris* gen. nov. sp. nov.,

the anterior lobe is largely effaced giving an inflated look to the front of the shields; this was likely to allow sufficient space within the bivalved carapace to hold the appendages of the animal.

Thus, the "tightly-enclosed" interpretation of bradoriids is likely correct, although adductor muscle scars have not been positively identified [28,29]. Cambrian arthropod fossils with conjoined bivalves that are opened out into the butterfly position may represent either shed exuviae (as occurs in *Tuzoia* and *Isoxys*) or situations where adductor musculature has experienced post-mortem loss [28].

Based on a survey of the preserved appendages that have been discovered so far, bradoriids never developed oar-like rowing appendages as in cladocerans such as *Daphnia* [30]. I propose here that the shields in some bradoriid genera were utilized for flap swimming, in addition to the protective function provided by closure of the two shields along a plane of commissure.

The depth of each shield in comparison to its width is large in several bradoriid genera, such as *Hipponicharion*, *Beyrichona*, and *Albrunnicola* (see genera descriptions below). The hydrodynamically-shaped wing tip (ventral valve edge) in *Beyrichona avannga* with its very smooth, rounded contour shows evidence for streamlining [31]. The appendages in bradoriids [16,17] show no modification for swimming efficiency, and thus the animal using a flap of its shields for rapid escape, as in scallops, seems plausible.

The delta-wing configuration of *Hipponicharion skovstedi* also suggests swimming by clapping of valves [31]. *Beyrichona papilio*, named for the butterfly in honor of its wing-like shields, has strongly acuminate (narrow) ventral shield margins and was perhaps able to "fly" through the water as may have been the case for other species in the genus *Beyrichona* with tall, triangular shields such as *Beyrichona tinea*.

The same is true, perhaps even more so, of the bradoriid *Pseudobeyrichona monile* with its wing-shaped shields [19]. *Matthoria troyensis* also has a postplete shield shape well suited to shield-rowing locomotion, with a bluntly pointed extension at the ventral most part of the marginal rim [20]. The preplete configuration of the shields of *Cambroarchilocus tigris* gen. nov. sp. nov. suggests that shield swimming might have propelled the animal backwards comparable to the motion of a jet-propelled cephalopod. A more appropriate analogy may be a rowing motion with the shields. This can also explain why the dorsoposterior lobe is so well delimited; the posterior lobes may have functioned (with carapace closed) something like a pair of stabilizing fins (*cf.* cephalopod fins) during jet propulsive backwards locomotion.

Sustained swimming by clapping of valves, as in the swimming file clam (*Limaria* spp.), is a locomotary mode that may have applied to bradoriids, especially the beyrichonids and hipponicharionids which developed particularly tall shields. The fragile file clam (*Limaria fragilis*) swims by opening its valves and then expelling water by closing its valves. It can swim in this fashion for five minutes or more [32]. The reconstruction of *Monceretia* in frontal view [33] seems poised for this type of underwater "flight".

## 5. Bradoriid Developmental Biology

A hypothetical allele of *Antennapedia*, *Antp$^{Tgl}$*, has been proposed as being responsible for [34] morphological changes in trilobites such as cephalic width changes and the development of greatly elongated pleural segments, the macropleurae [34,35]. Hughes [35] made an alternate proposal, linking the development of macropleural segments in trilobites to *Abdominal-B* Hox gene with expression being linked to the location of the genital opening, with its position along the animal's anterior-posterior axis being the proximal trigger for macropleural development [35]. Unfortunately for Hughes' *Abdominal-B* scheme for the development of macropleurae, if we roughly equate macropleural development with appendage development, and there exist good reasons to do so, *Abdominal-B* actually represses rather than expresses the development of appendages [36].

The good reasons to equate macropleural development with appendage development are as follows. The unusual terminal cerci in *Olenoides serratus* are antennae-like structures that might very well be considered ectopic antennae resulting from Hox gene expression. Said gene might very well be a gene from the *Antennapedia* complex such as $Antp^{Tgl}$. Interestingly, *Paradoxides grandoculus* [37] lacks cerci but develops a pair of final thoracic pleurae that are greatly elongated into cerci-like structures. Incidentally, both the cerci and the terminal thoracic macropleurae in *Olenoides serratus* and *Paradoxides grandoculus* may have had synecological significance for the animals by serving as an early warning system against attack from behind. Both structures, furthermore, may have been controlled by the $Antp^{Tgl}$ allele showing expression in the ventral part of the animal (cerci) or the dorsal (i.e., carapace) part of the animal (terminal thoracic macropleurae).

The shape or "swing" of bradoriid valves comes in three versions: preplete, amplete, and postplete (Figure 3). The terms refer to whether the maximum height of the shield is anterior (preplete), near the midline (amplete), or posterior (postplete). I propose here, in accordance with the $Antp^{Tgl}$ hypothesis, that the shape of the bradoriid shield was also controlled by the $Antp^{Tgl}$ allele in a way similar to the way that it influenced the shape of the trilobite cephalon. The influence of *Antennapedia* can help explain the otherwise hard to explain deep (tall) triangular to subtriangular shield shapes in the Beyrichonidae and the Hipponicharionidae. The maximum height of the shield (see Figure 3) in a particular bradoriid species, in the view advocated here, was controlled by the expression of the $Antp^{Tgl}$ allele, with the exact position of the expression on the gene along the anterior-posterior axis of the animal determining whether the shield is preplete, amplete, or postplete. For example, if the maximum shield height is anterior, the shield is proplete (Figure 3, left); if the maximum shield height is central, the shield is amplete (Figure 3, center); and if the maximum shield height is posterior, the shield is postplete (Figure 3, left). This corresponds to the allele's influence on the width of the cephalon in nevadiid and bristoliid trilobites [34]. The proximity of the posteroventral node to the ventral margin, e.g., adjacent as in *Shangsiella* versus more distant in *Petrianna* [33] may also be controlled by the $Antp^{Tgl}$ allele in the same way that interocular spacing varies from state 1 to 5 and extraocular spacing varies from state 1 to 4 in nevadiid trilobites [38].

Hughes [35] argued that "*Antennapedia* expression is largely confined to the trunk and posterior cephalic body regions among extant arthropods" and this would accord with its influence proposed here for bradoriid shield shape. Hughes [35], citing the work of his student Webster [39], mistakenly commented "the role of $Antp^{Tgl}$ in the development of cephalic morphology is inconsistent with current knowledge of olenellid ontogeny." Rather, if we accept that *Antennapedia* can influence the head region as well (as apparent in the eponymous *Drosophila* mutant [40]), the position of the genal and metagenal spines in *Nephrolenellus multinodus* through ontogeny is in accord with predictions of the $Antp^{Tgl}$ model, especially if you consider the enhanced width of the first glabellar lobe in mature individuals of this species.

## 6. Bradoriid Genera

Following is a review of accepted bradoriid genera (plus a few unnamed forms), with family assignment (if known) and geological ranges listed for each genus.

*Acutobalteus* Betts et al. 2017 [41]; Range: early Cambrian, *Dailyatia odyssei* Zone; Hipponicharionidae; In *Acutobalteus* valves, the two lobes merge distally, forming a single lobe [31].

*Albrunnicola* Martinsson, 1979 [42]; Hipponicharionidae; Range: Stage 3 to Drumian. This genus was initially given the name *Longispina* by D. Andres in 1969 [43]. In an amusingly-written short note published by Anders Martinsson in 1979 [42], this name was shown to have a senior homonym, a brachiopod from the Middle Devonian. Martinsson thus introduced the replacement name *Albrunnicola*. According to Martinsson [42], the "name is derived from the root *albrunn-* in the name of the type locality on the Isle of Öland and *-cola* (Latin), inhabitant. Note particularly that the gender is masculine (hence *Albrunnicola oelandicus*) and that the short penultimate syllable places the stress on the connecting vowel *i*." In this genus, the valves are as tall or taller than the hinge line is

long. A thin flat margin runs from one edge of the hinge to the other. Lobes may slightly overhang the hinge area. *Albrunnicola* (Figure 4) "is widespread hipponicharionid genus characterized by a strongly developed anterodorsal lobe and a weaker, node-like posterodorsal lobe" [6]. *Albrunnicola* is distinguished from other hipponicharionids by "a weakly developed posterior lobe, with both lobes restricted to the dorsal half of the valve" [25]. *A. bengtsoni* from the Wirrapowie Limestone in the Flinders Ranges, Australia is a "weakly postplete bradoriid with subtriangular lateral outline" [6]. The genus occurs in the Cambrian Series 2 Bastion Formation of Greenland [31]. *Albrunnicola bengtsoni* (Figure 4) has been reported from glacial erratic clasts from the Transantarctic Mountains [44]. The type species of the genus is middle Cambrian (Cambrian Series 3, Wuliuan; *Albrunnicola oelandicus* [43]), and occurs in Öland, Sweden, in the Borgholm Formation [31]. Interestingly, the subspecies *Albrunnicola oelandicus maroccanus* Hinz-Schallreuter, 1993 has been reported from the central Anti-Atlas of Morocco [19]. *Albrunnicola* has an interesting paleobiogeographic distribution, occurring in Laurentia, Australia, and Antarctica [31]. The genus is synonymous with *Beyrichona chinensis* [25].

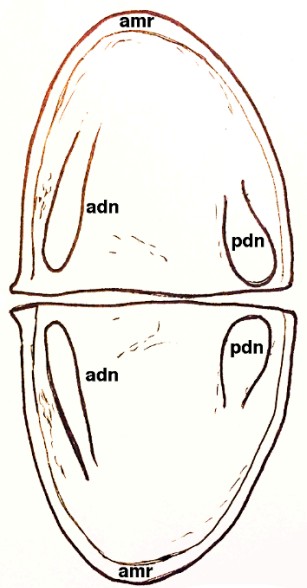

**Figure 4.** *Albrunnicola bengtsoni*. Paired shields shown in butterfly position. Anterior to left. Length of shield 1.5 mm. Abbreviations: **adn**, anterodorsal node; **amr**, admarginal rim; **pdn**, posterodorsal node. Image Credit: Mark McMenamin.

*Almazina* Melnikova, 2003 [45]; Range: Cambrian. *Almazina*, from Malyi Karatau, Russia, differs from *Euzepaera* by having smaller, more densely distributed pores [6].

*Altajanella* Melnikova, 1992 [46]; Range: Late Pabian to early Stage 9. Earlier thought to have true ostracod affinities [5], this tiny bradoriid is now thought to be homoplasic on ostracod "body plan 2" [2].

*Aluta* Matthew, 1896 [47]; Alutidae; Range: Early Cambrian, Series 2. *Aluta* differs from *Bradoria* by the presence of a broad marginal rim of equal width from anterior to posterior [21]. *Aluta* occurs in the Niutitang Formation of the Yangtze Platform of Guizhou, China.

*Alutella* Kobayashi & Kato 1951 [48]; Comptalutidae; Range: Middle Stage 3 to mid Wuliuan. *Alutella* has been reported from Scania, Sweden [48].

*Amphikeropsis* Topper, Skovsted, Brock and Paterson, 2007 [25]; Range: Late Stage 3. This genus is endemic to South Australia [25].

*Anabarochilina* Abushik, 1960 [49]; Svealutidae; Range: Late Stage 3 (*Elliptocephala asaphoides* fauna, New York) to late Stage 10. *Anabarochilina* is an oval-elongate, postplete to subamplete, smooth bradoriid with a fairly wide marginal rim, a well-developed hinge line and up to three nodes [20]. A partially exfoliated valve of *Anabarochilina* sp. from the Middle Cambrian of Kazakhstan shows what are interpreted as integumental hemolymph networks, as does *Anabarochilina primordialis* from the middle Cambrian of Västergötland, Sweden [2]. The genus represents a cosmopolitan, presumably pelagic taxon [5,50]. A single prominent node per valve is present in the cosmopolitan species *Anabarochilina australis* [51]. *Anabarochilina primordialis* may possess an enlarged brood chamber. One species of the genus may have lived at southern hemisphere high latitude (70° S) sea surface habitat in the Middle Cambrian (*Lejopyge laevigata* Biozone) [5].

*Annge* Smith et al. 2014 [52]; Range: Cambrian Series 3, Stage 5. *Annge iperte* occurs in the Giles Creek Dolostone in Australia.

*Antihipponicharion* Huo & Shu, 1985 [53]; Indianidae; Range: Cambrian.

*Aristaluta* Öpik, 1961 [54]; Comptalutidae; Range: Late Wuliuan to mid-Guzhangian.

*Auriculatella* Tan, 1980 [55]; Cambriidae; Range: Mid to late part of early Stage 3. *Auriculatella* is distinguished by a posterodorsal ridge that connects to the anterodorsal ridge by means of a connecting ridge, and also by presence of an anterodorsal node [38]. *Auriculatella typica* occurs in the *Abadiella* Biozone of South China [5].

*Australopsis* Hinz-Schallreuter, 1993 [56]; Range: Late Wuliuan.

*Beyrichona* Matthew, 1886 [57]; Beyrichonidae; Range: Late Stage 3 to Ordovician (Tremadoc). *Beyrichona* was first described from Avalonia by Matthew in 1886 [57]. The genus is known for its subtriangular carapace [7]. *Beyrichona* is a characteristic genus (Figures 5–7) of the Cambrian Series 2–3 Hanford Brook Formation of New Brunswick [31]. It has also been assigned to *Neokunmingella* [6]. *Beyrichona* is considered to be characteristic of shallow water deposits [31]. *Beyrichona avannga* (Figure 5) can be hard to distinguish from some species of *Albrunnicola* [31]. *Beyrichona* is a very long ranging bradoriid, surviving into the Ordovician until going extinct at the end of the Tremadocian (*Angelina sedgwickii* Biozone) [2].

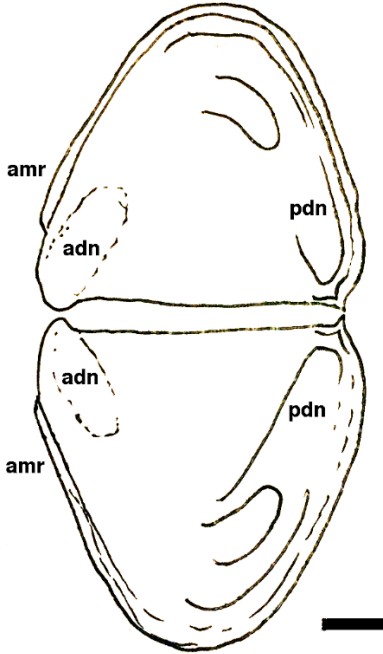

**Figure 5.** *Beyrichona avanga*. Paired shields shown in butterfly position. Anterior to left. Scale bar 200 microns. Abbreviations: **adn**, anterodorsal node; **amr**, admarginal rim; **pdn**, posterodorsal node. Image Credit: Mark McMenamin.

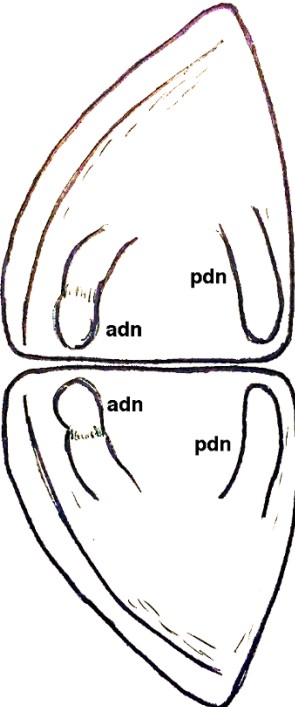

**Figure 6.** *Beyrichona papilio*. Paired shields shown in butterfly position. Anterior to left. Length of shield 3.6 mm. Abbreviations: **adn**, anterodorsal node; **pdn**, posterodorsal node. Image Credit: Mark McMenamin.

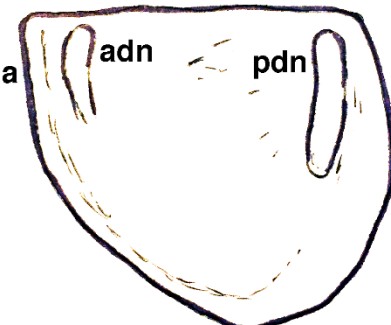

**Figure 7.** *Beyrichona tinea*. Anterior to left. Length of shield 3.2 mm. Abbreviations: **a,** anterodorsal cardinal marginal concavity; **adn**, anterodorsal node; **pdn**, posterodorsal node. Image Credit: Mark McMenamin.

*Biaurina* Hinz-Schallreuter, 1993 [56]; ?Bradoriidae [58]; Range: Late Wuliuan. This genus has been reported to have a doublure, in which case it would be classified as a phosphatocopid, but the presence of a doublure is doubted [6], so it is included here as a bradoriid based on a lobation pattern that matches that commonly seen in Bradoriida. Surface ornamentation in *Biaurina punctata* consists of pores open to the exterior [6].

*Bicarinellata* Stigall, 2008 [59]; Hipponicharionidae; Range: Late Stage 3 to early Stage 4 (with question). *Bicarinellata* is a replacement name for *Bicarinella* Rode, Lieberman and Rowell, 2003 which was preoccupied by two (*sic*) gastropod genus names. *Bicarinellata evansi* (Figure 8) occurs in lower Cambrian strata of the Pensacola Mountains, East Antarctica [6,24]. The posterior lobe or node in *Bicarinellata* is close to the edge of the valve [31].

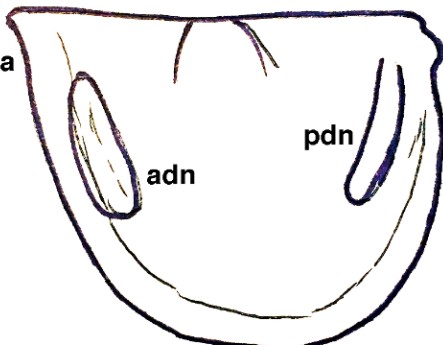

**Figure 8.** *Bicarinellata evansi.* Anterior to left. Letter "a" denotes anterodorsal cardinal corner marginal concavity. Length of shield 1.5 mm. Abbreviations: **a,** anterodorsal cardinal marginal concavity; **adn**, anterodorsal node; **pdn**, posterodorsal node. Image Credit: Mark McMenamin.

*Bradoria* Matthew, 1899 [60] (Figure 9); Bradoriidae; Range: Late Stage 3 to early Wuliuan; mid Wuliuan to late Stage 9 (with question). Surface ornamentation in *Bradoria scrutator* consists of pores that do not penetrate the shell and can only be seen where the shell is exfoliated [6].

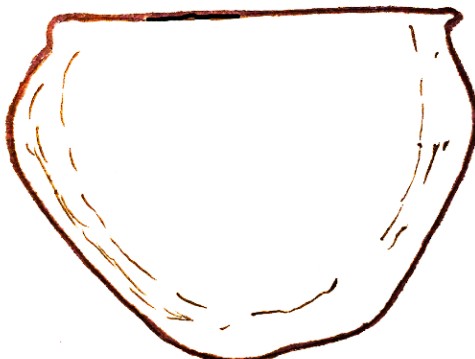

**Figure 9.** *Bradoria* sp. Anterior to left. Length of shield 1.28 mm. Image Credit: Mark McMenamin.

*Bullaluta* Copeland, 1986 [61]; Bradoriidae; Range: Late Guzhangian. *Bullaluta* is a western Newfoundland bradoriid of Furongian (Late Cambrian) age [19]. It has a non-lobate shield [19], with a pustulose appearance where the valve is exfoliated [20]. Like Hipponicharionidae, the Beyrichonidae have a subtriangular shield [25]. The anterior and posterior lobes extend further ventrally in the Hipponicharionidae than in the Beyrichonidae. Posterior sulcation often occurs in the Beyrichonidae [25].

*Cambria* Neckaja & Ivanova 1956 [62]; Cambriidae; Range: Stage 3. *Cambria* (Figures 10 and 11) has a large carapace with an inflected hinge line [7]. The terminal lobes are divided by a sulcus, and a papillate surface may be present [7]. The anterodorsal node is linked to the posterodorsal node by a connecting ridge; an anterodorsal ridge is present [33]. Formerly assigned with question to the Hipponicharionidae [7], it is now the eponymous genus for the Cambriidae [25]. The Montagne Noir (Pardailhan Formation) of southern France has produced *Cambria danvizcainia* [19,33]. *Cambria melnikovi* (Figure 10) from the Atdabanian of eastern Siberia shows what are interpreted as integumental hemolymph networks [2].

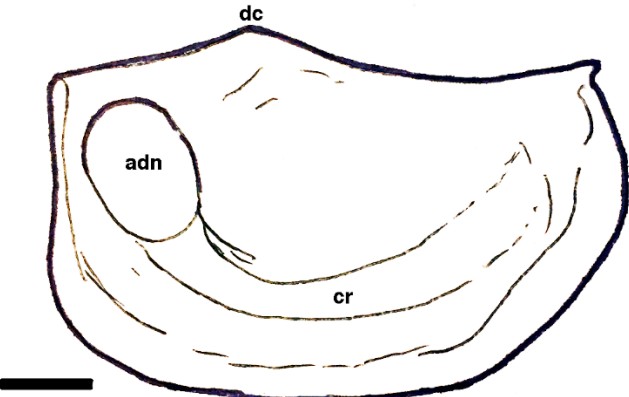

**Figure 10.** *Cambria melnikovi*. Anterior to left. Scale bar 1 mm. Abbreviations: **adn**, anterodorsal node; **cr**, connecting ridge; **dc**, dorsal cusp. Image Credit: Mark McMenamin.

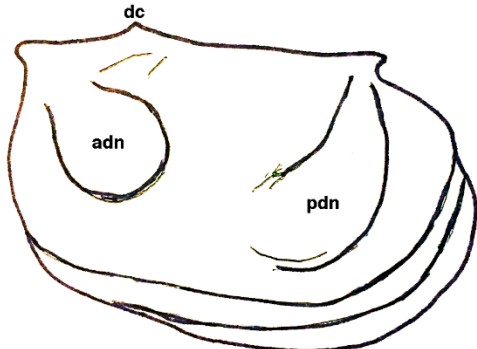

**Figure 11.** *Cambria sibirica*. Anterior to left. Length of shield 3 mm. Abbreviations: **adn**, anterodorsal node; **dc**, dorsal cusp; **pdn**, posterodorsal node. Image Credit: Mark McMenamin.

*Cambroarchilocus* nov. gen.; Hipponicharionidae; Range: Late Stage 3. See new genus description below.

*Capricambria* Hinz, 1991 [63]; Monasteriidae; Range: Late Wuliuan. A spiny bradoriid.

*Cedocamia* Hinz-Schallreuter, 1993 [56]; Comptalutidae; Range: Late Wuliuan.

*Chegetella* Kanygin, 1977 [64]; Range: middle Ordovician. This spinose bradoriid occurs in Siberian middle Ordovician strata [6,13]. *Chegetella chegitunica* is the correct species spelling, not "C. chegetunica".

*Chuanbeiella* Huo & Peng (in Huo & Shu), 1985 [53]; Cambriidae; Range: Cambrian.

*Comptaluta* Öpik, 1968 [65]; Comptalutidae; Range: Middle Stage 3 to Drumian. *Comptaluta* consists of small, decidedly postplete bradoriids; the submedian node is prominent [21]. *Comptaluta kailiensis* and *C. inflata* occur in the Balang Formation of Zhenyuan County, Guizhou Province, South China [21].

*Cordubiella* Hinz-Schallreuter, Gozalo & Liñán, 2008 [50]; Cambriidae; Range: Stage 3, early. *Cordubiella pedrochensis* (Figure 12) of southern Spain [50] (*Lemdadella perejoni* Biozone, late Issendalenian, below the *Eoredlichia* trilobite first appearance datum) is one of the oldest bradoriids from Western Gondwana [19]. In *Cordubiella pedrochensis*, the nodular anterior lobe is "located at inner margin of anterodorsal field, posterior lobe ridge-like and extending from posterodorsal to central field. Connecting lobe distinct" [50]. These lobes are more pronounced than in *Cambria*. The carapace in this species reaches to 7 mm in length.

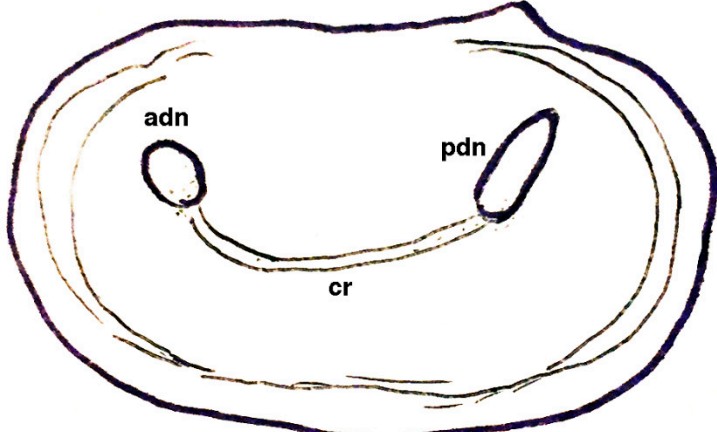

**Figure 12.** *Cordubiella pedrochensis*. Anterior to left. Length of shield 5.1 mm. Abbreviations: **adn**, anterodorsal node; **cr**, connecting ridge; **pdn**, posterodorsal node. Image Credit: Mark McMenamin.

*Dictyocharion* Jones & Kruse 2009 [58]; Hipponicharionidae; Range: Middle Cambrian. A subtriangular bradoriid with a coarse polygonal valve ornamentation consisting of rounded ridges from the Thorntonia Limestone of the Georgina Basin, Northern Territory, Australia [58].

*Dielymella* Ulrich and Bassler, 1931 [66]; family unknown [5]; Range: Wuliuan, with question Stages 3, 4, and Drumian to late Stage 9. *Dielymella's* large preplete, punctate shields are elongate. A short anterior spine is present [20]. *Dielymella?* occurs in the Montezuman Stage (Cambrian Stage 3) in the Montezuma Range and Indian Springs Canyon sections in Nevada, USA [67], and a comparable specimen occurs in Sonora, México (see below).

*Dietmarandresia* Hinz-Schallreuter, 2008 [68]; Hipponicharionidae; Range: Middle to late Drumian (with question). This is a new name for *Andresia* Hinz-Schallreuter, 1993, a name preoccupied by *Andresia* Stephenson, 1921 [69]. This genus is known for its "confluent anterior and posterior lobes" [25].

*Dorispina* Hinz-Schallreuter, 1993 [56]; family uncertain [58]; Range: Late Wuliuan.

*Dubianella* Shu, 1990 [70]; Duibianellidae; Range: Middle Stage 3; with question to early Drumian. The valves of *Dubianella* are covered in nodes, giving valves an irregular lumpy aspect that is very reminiscent of one variant of the Belgian Oligocene ostracod *Cyprideis* (*Goerlichia*) *williamsoniana* [71]. In both cases, we might ask if some type of parasite or endobiont has triggered deformity in valve surface morphology. A few spines may also be present, and these may be covered by a scale-like ornamentation as seen in the spines of *Mongolitubulus*. *Dubianella* is known from both Siberia and China. As *Dubianella* resembles juveniles of *Spinospitella* [6], it is inferred here to be a neotenous taxon.

*Emeiella* Li, 1975 [72]; Kunmingellidae; Range: Mid to late part of early Stage 3. This kunmingellid (*Emeiella* sp.) has been reported from the Jince Formation of the Czech Republic [19]. One of the earliest bradoriids, *Emeiella venusta* occurs in the *Abadiella* Biozone of South China [5].

*Eozhexiella* Betts et al. 2017 [41]; Range: early Cambrian, *Dailyatia odyssei* Zone. *Eozhexiella adnyamathanha* is an Australian species.

*Epactridion* Bengtson in Bengtson et al. 1990 [73]; Monasteriidae; Range: Stage 3, early. *Epactridion portax* has punctae that open to the exterior [6]. The punctae in this species are particularly prominent in early instars [73]. Bengtson [73] interpreted the end of *Epactridion portax* with projecting spines as its anterior (also rendering it a postplete form). The anterior versus posterior ends of bivalved arthropod shields can be difficult to accurately ascertain [7], but Bengtson's [73] interpretation is likely correct considering the slight concavity in the shield margin between the cardinal corner and spine (the anterodorsal cardinal marginal concavity).

*Eremos* Hinz-Schallreuter, 1993 [74]; Svealutidae; A cosmopolitan, presumably pelagic taxon [50]. *Eremos australis* is synonymous with *Anabarochilina australis* [51].

*Euzepaera* Shu, 1990 [70]; Zhexiellidae; Range: Stage 3; with question to Furongian late Stage 9. *Euzepaera* has a punctate shell with the pores sealed on the inside by domal pads [6]. The pad domes have flattened tops, implying that the valve had an inner layer surrounding the pads that was not preserved. The genus is distinguished by the absence of easily discerned ridges or lobes, but a V-shaped anterodorsal sulcus is present [6]. *Euzepaera* sp. (Figure 13) occurs in the Wirrapowie Limestone in the Flinders Ranges, Australia [28].

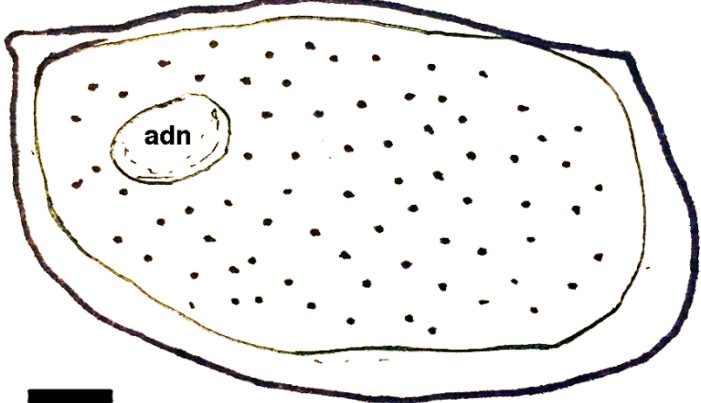

**Figure 13.** *Euzepaera* sp. Anterior to left. Scale bar 100 microns. Abbreviation: **adn**, anterodorsal node. Image Credit: Mark McMenamin.

*Flemingopsis* Jones & McKenzie, 1981 [75]; Range: Late Wuliuan to Drumian. *Flemingopsis* is a replacement name for *Flemingia*. This bradoriid has a morphologically simple shield and could be confused with *Houlongdongella*, *Zepaera* and *Quetopsis* [25].

*Flumenoglacies* Peel & Streng 2015 [76]; Hipponicharionidae; Range: Cambrian Series 3. *Flumenoglacies* is a hipponicharionid bradoriid from British Columbia and North Greenland from Cambrian Series 3 [31]. Lobation in *Flumenoglacies* develops a single comarginal lobe [31].

*Galeopsis* Hinz-Schallreuter, 1993 [56]; ?Beyrichonidae; Range: Late Wuliuan.

*Gladioscutum* Hinz-Schallreuter & Jones, 1994 [77]; Cambriidae; Range: Middle Cambrian, Late Drumian. *Gladioscutum* is considered to be a late surviving cambriid taxon [5].

*Hanchiangella* Huo et al. 2002 [4]; Kunmingellidae; Range: early Cambrian *Abadiella* Biozone. One of the earliest bradoriids, *Hanchiangella minor* (Figure 14) occurs in the *Abadiella* Biozone of South China [5].

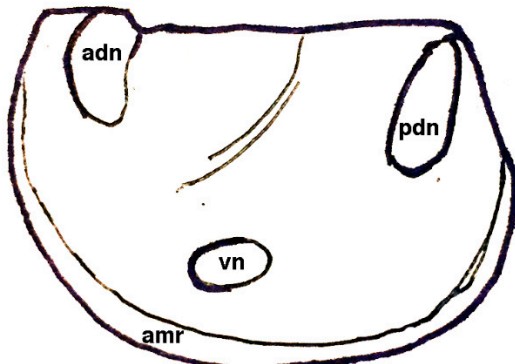

**Figure 14.** *Hanchiangella minor*. Anterior to left. Length of shield approximately 3.5 mm. Abbreviations: **adn**, anterodorsal node; **amr**, admarginal rim; **pdn**, posterodorsal node; **vn**, ventral node. Image Credit: Mark McMenamin.

*Hanchungella* Huo, 1956 [78]; Bradoriidae; Range: Stage 3, early. *Hanchungella* has a non-lobate shield [19].

*Haoia* Shu, 1990 [70]; ?Haoiidae; Range: Middle to late Stage 3. *Haoia* cf. *H. shaanxiensis* from the Mernmerna Formation, Flinders Ranges, South Australia has a coarse ornament of elongate ridges with small pustules in the low spaces between ridges [6]. The genus is distinguished by consisting of "weakly postplete bradoriids with a V-shaped anterodorsal sulcus and multiple, ridge-like lobes" [6].

*Hipponicharion* Matthew, 1886 [57]; Hipponicharionidae; Range: Stage 3 to Wuliuan. *Hipponicharion* (Figures 15 and 16) was first described from Avalonia by Matthew in 1886 [57]. It is a characteristic genus of the Cambrian Series 2–3 Hanford Brook Formation of New Brunswick [31]. *Hipponicharion geyeri* has punctae that open to the exterior [6]. The posterior lobe in *Hipponicharion* is close to the edge of the valve [31]. *Hipponicharion* is considered to be characteristic of shallow water deposits, and is common to both east and particularly west Gondwana [19], while also occurring in Laurentia and South China [31]. A growth series of *Hipponicharion loculatum* (Figure 15) shows isometric growth [43]. Hipponicharionids are characterized by a clear marginal rim, a dorsal central node, and posterior and anterior lobes that are not connected [24]. Similar to the Beyrichonidae, they have a deep subtriangular shield. Hipponicharionid genera are distinguished by "shield shape, size and ventral extent of the anterior and posterior lobes" [25] and by "a straight dorsal margin, a well-defined hinge-line, a narrow, entire admarginal ridge and prominent dorsoventrally orientated lobation in which the anteriormost and posteriormost lobes are subparallel to the adjacent ... valve outline" [20].

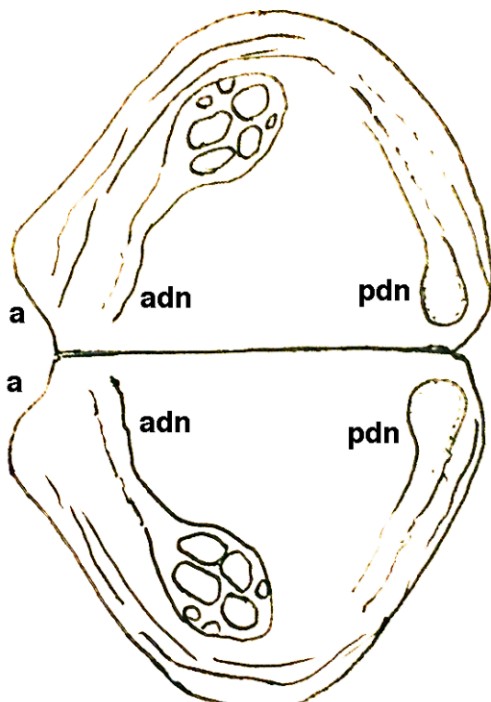

**Figure 15.** *Hipponicharion loculatum*. Paired shields shown in butterfly position. Anterior to left. Length of shield 1.5 mm. Abbreviations: **a,** anterodorsal cardinal marginal concavity; **adn**, anterodorsal node; **pdn**, posterodorsal node. Image Credit: Mark McMenamin.

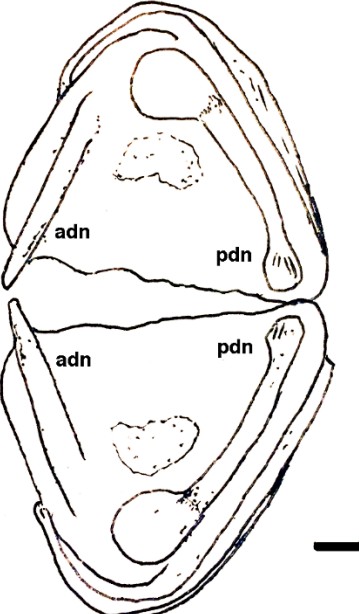

**Figure 16.** *Hipponicharion skovstedi*. Paired shields shown in butterfly position. Anterior to left. Scale bar 200 microns. Abbreviations: **adn**, anterodorsal node; **pdn**, posterodorsal node. Image Credit: Mark McMenamin.

*Houlongdongella* Li, 1975 [72]; Zhexiellidae; Range: Middle Stage 3. This bradoriid has a morphologically simple shield and could be confused with *Flemingopsis*, *Zepaera* and *Quetopsis* [25].

*Ifersiktia* Hinz-Schallreuter, 1993 [74]; Bradoriidae; Range: early Wuliuan. This bradoriid, *Ifersiktia mipora*, is from Jbel bou Ifersikht, Anti-Atlas, Morocco [19,74].

*Indiana* Matthew, 1902 [79]; Bradoriidae; Range: Stage 3 to early Wuliuan. The valves of *Indiana* are smooth and relatively featureless, although some species may show fine punctae [7]. Once assigned to the family Indianidae [7], it was later transferred to the Bradoriidae [25]. Specimens of *Indiana lentiformis* (Figure 17) reaching up to 13.3 mm in length occur in the Comley Series of Wales and England [20]. *Indiana* aff. *I. dermatoides* and *Indiana* sp. have been reported from the Franconian Forest of eastern Bavaria, Germany [19]. *Indiana* is characteristic of shallow marine environments [19].

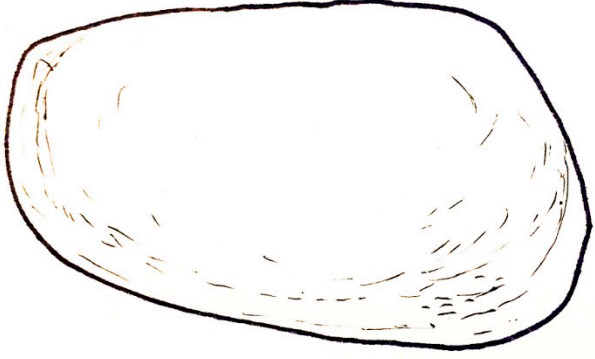

**Figure 17.** *Indiana lentiformis*. Anterior to left. Length of shield 13.3 mm. Image Credit: Mark McMenamin.

*Indianites* Ulrich & Bassler, 1931 [80]; Bradoriidae; *Indianites caenensis* was the first bradoriid discovered in France, from the Saint-jean-de-la-Rivière Formation, Carteret, Normandy (*Bigotina* Biozone) [19]. The species may in fact be a simple svealutid [19].

*Indota* Öpik, 1968 [65]; Bradoriidae; Range: Stage 3 to early Wuliuan. *Indota*? sp. has been reported from the Franconian Forest of eastern Bavaria, Germany [19].

*Jiucunella* Hou & Bergström, 1991 [81]; ?Alutidae; Range: Middle Stage 3. *Jiucunella* occurs in the Wirrapowie Limestone in the *M. etheridgei* Zone, Flinders Ranges, Australia [28,82]. *Jiucunella phaseloa* Figure 18 has a valve outline (tilted forward) similar to that of *Jixinlingella daimonikoa*, but has a largely smooth valve surface and lacks the polygonal ornamentation.

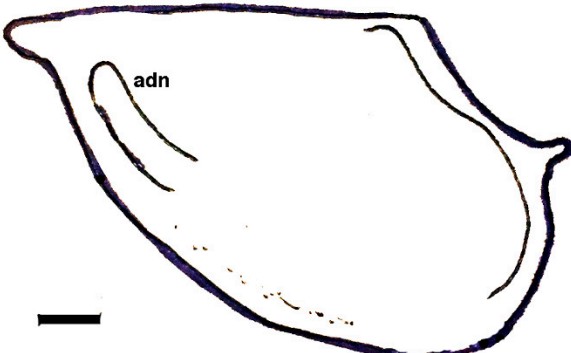

**Figure 18.** *Jiucunella phaseloa*. Anterior to left. Scale bar 200 microns. Abbreviation: **adn**, anterodorsal node. Image Credit: Mark McMenamin.

*Jixinlingella* Li, 1975 [72]; ?Alutidae; Range: Middle Stage 3. *Jixinlingella* occurs in the Wirrapowie Limestone in the *M. etheridgei* Zone, Flinders Ranges, Australia [28,82]. *Jixinlingella daimonikoa* is covered by a coarse polygonal ornament of ridges forming a pattern of interlocked hexagons (giving the valves a surface texture that somewhat resembles the surface texture of a roasted peanut shell), and a two-pointed spine projects from the posterior of the valve. The valve ornamentation is comparable to that of the ostracod *Cythereis cretaria dorsocaudata* from Late Cretaceous strata of Israel [83]. Short denticulate spines occur along the ventral margin of the valve in *Jixinlingella daimonikoa* (Figure 19). The ridge ornament is comparable to the ridge ornament of the Eocene-Oligocene Belgian ostracod *Quadracythere macropora* [71]. This genus is sometimes misspelled "Jixinglingella" [2].

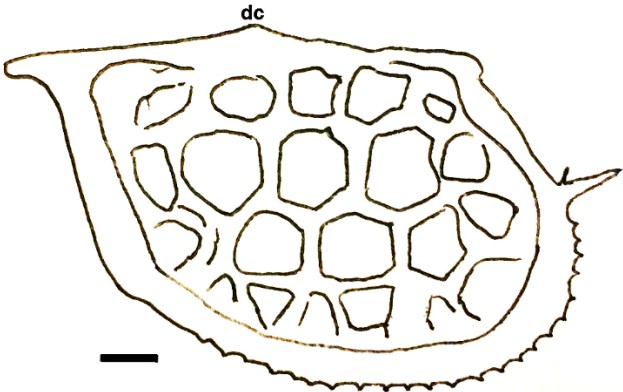

**Figure 19.** *Jixinlingella daimonikoa*. Anterior to left. Scale bar 200 microns. Abbreviation: **dc**, dorsal cusp. Image Credit: Mark McMenamin.

*Kazakhstanotubulus* Gridina, 1991 [84]; Duibianellidae; This genus, from the lower Cambrian of central Kazakhstan [6], may represent a bradoriid spine. Second-order spines are present. The short curved spine *Seletellus* can co-occur with *Kazakhstanotubulus* and the two may represent parts of the same animal [6].

*Koksodus* Missarzhevsky & Mambetov, 1981 [85]; *Koksodus* is a spiny possible bradoriid fragment from lower Cambrian strata of Malyi Karatau [6].

*Konicekion* Šnajdr, 1975 [86]; Hipponicharionidae; Range: Stage 3, early to middle. *Konicekion snajdri*, *K. tix*, and *Konicekion radion* all occur in the Jince Formation (middle Cambrian), Czech Republic. However, considering taphonomic issues and the morphological variability common to bradoriids, the three species of *Konicekion* may in fact represent a single species [19]. *Konicekion* may be synonymous with *Albrunnicola* [19].

*Kunmingella* Huo, 1956 [78]; Kunmingellidae; Range: Middle Stage 3 to earliest Stage 4. Due to its occurrence in the Chengjiang Lagerstätte, *Kunmingella douvillei* is the only bradoriid for which the limb morphology is known [3]. The anterior of the two lobes in *Kunmingella* (Figure 20) is equidimensional and short [31]. *Kunmingella* (along with *Kunyangella*) was recovered as the basal-most mandibulate stem in a phylogenetic analysis of the Panarthropoda [87]. A specimen of *Kunmingella douvillei* from the Maotianshan Shale shows what are interpreted as integumental hemolymph networks [2].

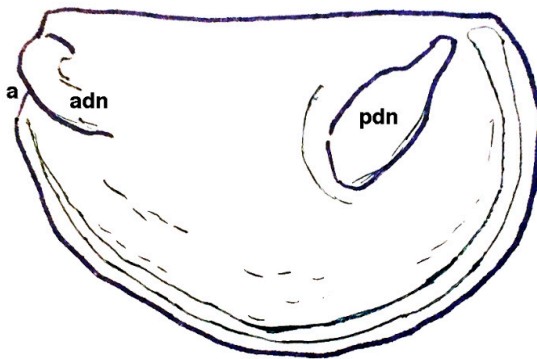

**Figure 20.** *Kunmingella maotianshanensis*. Anterior to left. Length of shield 4 mm. Abbreviations: **a,** anterodorsal cardinal marginal concavity; **adn**, anterodorsal node; **pdn**, posterodorsal node. Image Credit: Mark McMenamin.

*Kunyangella* Huo, 1965 [88]; Comptalutidae; Range: Middle Stage 3 to earliest Stage 4. *Kunyangella* (along with *Kunmingella*) was recovered as the basal-most mandibulate stem in a phylogenetic analysis of the Panarthropoda [87].

*Leperdita* Jones, 1856 [89]; This British bradoriid, *Leperdita solvensis*, was the one of the first to be described, and was called at the time an ostracod crustacean [89]. The genus is also referred to as *Leperditia* Rouault [90]. Note that Rouault [90] described a leperditicopid ostracod that was apparently convergent on Cambro-Ordovician leperditid bradoriids. Nevertheless, *Leperditia hicksi* is reported from the Middle Cambrian of South Wales [91] making it either a bradoriid or the oldest known ostracod. Walcott [27] illustrated what he described as *Leperditia dermatoides* from the Taconic strata of Washington County, New York. The species (greatest length 6 mm) is notable for roughly ellipsoidal shape, its finely punctate shield surface (which led Walcott [27] to consider assigning the species to a new genus), and a faintly marked, arched hinge line.

*Liangshanella* Huo, 1956 [78]; Svealutidae; Range: Stage 3 to early Drumian, with question to Furongian, late Stage 9. *Liangshanella* (Figure 21) consists of "postplete bradoriid valves from the Cambrian that display little or no lobation" [19]. The genus can be difficult to distinguish from *Indiana*, *Hanchungella*, *Ovaluta* and *Indota* [19]. *Liangshanella circumbolina* is a relatively smooth-walled species at first glance, but on closer inspection it bears surface pores with complex ornament (Figure 22) consisting of a ring divided into a dozen wedges and a central pit [25]. This feature is unknown from any other bradoriid, but might be compared to openings for sensory setae as occur in modern ostracods [25], as for example in the modern ostracod *Elofsonia baltica* [92]. These remarkable structures are only 5 microns in diameter. Protruding tubular structures with "sealed ends" occur on valve interiors, and resemble similar structures (domal pads capping interior protrusions) in *Euzepaera*. *Liangshanella* aff. *L. liangshanensis* and *Liangshanella* sp. have been reported from the Jince Formation of the Czech Republic [19]. *Liangshanella circumbolina* (Figures 21 and 22) occurs below the oldest trilobites (*Abadiella*

*huoi* Zone) in the Ajax Limestone, Flinders Ranges, Australia, and is thus the oldest known bradoriid (lower Cambrian, Series 2, Stage 3) from East Gondwana [25].

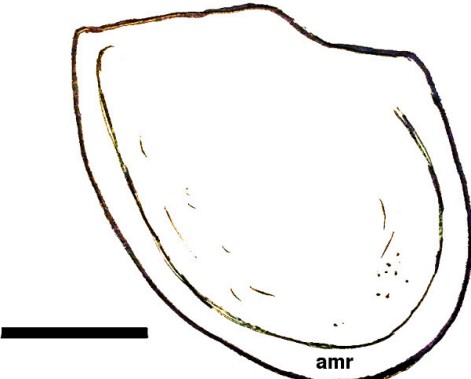

**Figure 21.** *Liangshanella circumbolina*. Anterior to left. Scale bar 500 microns. Abbreviation: **amr**, admarginal rim. Image Credit: Mark McMenamin.

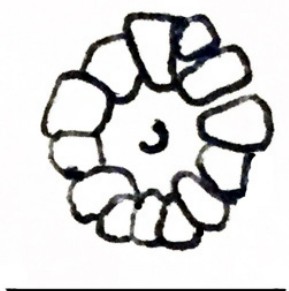

**Figure 22.** *Liangshanella circumbolina*. Detail of surface pore consisting of a ring divided into a dozen wedges with a central pit. Scale bar 5 microns. Image Credit: Mark McMenamin.

*Manawarra* Betts et al. 2017 [41]; Range: early Cambrian, *Dailyatia odyssei* Zone. *Manawarra jonesi* is an Australian bradoriid.

*Matthoria* Siveter & Williams, 1997 [20]; Cambriidae; Range: Late Stage 3 to early Stage 4. *Matthoria* has postplete valves with decided swing, and an anterodoral node that narrows towards the cardinal corner [33]. A posterodorsal node is lacking. The Montagne Noir (Pardailhan Formation) of southern France has produced ?*Matthoria* sp. [19,33]. This genus has obsolete lobation [5].

*Meishucunella* Jiang in Luo et al. 1982 [93]; Hipponicharionidae; Range: Stage 3, early. Lobation in *Meishucunella* develops a single comarginal lobe [31].

*Melvillella* Wrona, 2009 [44]; Svealutidae; Range: Late Stage 3 (with question).

*Monasterium* Melnikova in Melnikova et al. 1997 [94]; Monasteriidae; Range: Late Wuliuan to late Stage 9. An anterodorsally spiny bradoriid with an anterodorsal sulcus and reticulate ornamentation [5].

*Monceretia* Vannier, Williams, Álvaro, J.J., Vaizcaïno, D., Monceret, S., and Monceret, E., 2005 [33]; Cambriidae; Range: Late Stage 3. Postplete *Monceretia* has a posteroventral node but no other nodes; a short spine projects from the posterior cardinal corner [33]. The Montagne Noir (Pardailhan Formation) of southern France has produced *Monceretia erisylvia* [19].

*Mongolitubulus* Missarzhevsky, 1977 [95]; Duibianellidae; Range: Stage 3. This taxon, often preserved as an isolated, ornamented curved spine, has been the subject of much debate, with *Mongolitubulus* having been classified as a protoconodont, a pseudoconodont, a hyolithelminthid worm tube, and the defensive spine of a lobopodian. Melnikova [96] proposed the correct interpretation, namely, that these curved spines (here called *great spines*) represented prominent (to say the least) ornamentation projecting from the central parts of bradoriid valves. This interpretation was eventually confirmed by the discovery of a complete valve with its impressive spine still attached [97]. A defensive

function was implied [6]; however, a hydrodynamic function of these spines as stabilizers should also be considered. Considering their relatively fragile nature, these spines might be comparable in function to the spines of planktonic foraminifera. *Mongolitubulus henrikseni* has an array of both large and small spines. *Mongolitubulus anthelios* (Figure 23) occurs in the Wirrapowie Limestone in the *M. etheridgei* Zone, Flinders Ranges, Australia [82]. *Mongolitubulus* sp. occurs in the Wirrapowie Limestone in the Flinders Ranges, Australia [82]. The prominent lateral spines on *Mongolitubulus* are reminiscent of the prominent valve-central spines of the Eocene-Oligocene Belgian ostracod *Cytheropteron gulincki* [71].

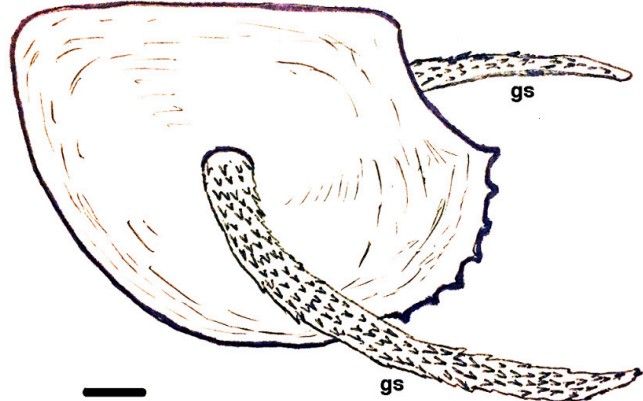

**Figure 23.** *Mongolitubulus anthelios*. Oblique side view, anterior to left. Scale bar 200 microns. Abbreviation: **gs**, great spine. Image Credit: Mark McMenamin.

*Mononotella* Ulrich and Bassler, 1931 [66]; Range: Late stage 4. *Mononotella* is a univalve bradoriid that lacks lobation and a lateroadmarginal rim [5].

*Nanchengella* Huo, 1956 [78]; Kunmingellidae; Range: Stage 3, early. *Nanchengella acuta* occurs in the *Abadiella* Biozone of South China [5].

*Navarana* Peel, 2017 [31]; Hipponicharionidae; *Navarana pearylandica* (Figure 24) occurs in the *Ovatoryctocara granuata* assemblage of the Henson Gletscher Formation, at a stratigraphic position at the top of Cambrian Series 2 [31]. Due to its prominent lobes (two each on the lateral sides of each valve), *Navarana* is placed with the hipponicharionids. The posterior lobe in *Navarana* is close to the center of the valve [31]. The lobes are ornamented with prominent tubercles [31]. Lobe separation distinguishes *Navarana* from other genera.

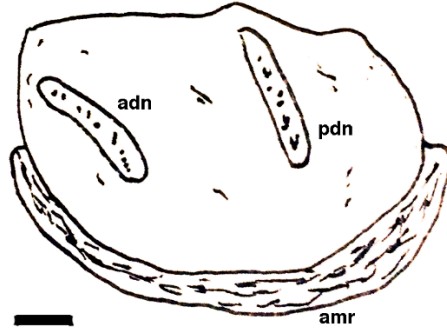

**Figure 24.** *Navarana pearylandica*. Anterior to left. Scale bar 100 microns. Abbreviations: **adn**, anterodorsal node; **amr**, admarginal rim; **pdn**, posterodorsal node. Image Credit: Mark McMenamin.

*Neoduibianella* Shu, 1990 [70]; Duibianellidae; Range: Middle Stage 3; with question to early Drumian.

*Neokunmingella* Zhang, 1974 [98]; Hipponicharionidae; Range: Middle Stage 3 to Wuliuan; with question to Tremadoc (Ordovician). Lobation in *Neokunmingella* (Figure 25) develops a single

comarginal lobe [31]. *Neokunmingella moroensis* occurs in the Wirrapowie Limestone in the Flinders Ranges, Australia [28,82]. This genus is known for sometimes showing "confluent anterior and posterior lobes" [25]. Each shield has a distinct posterior spine.

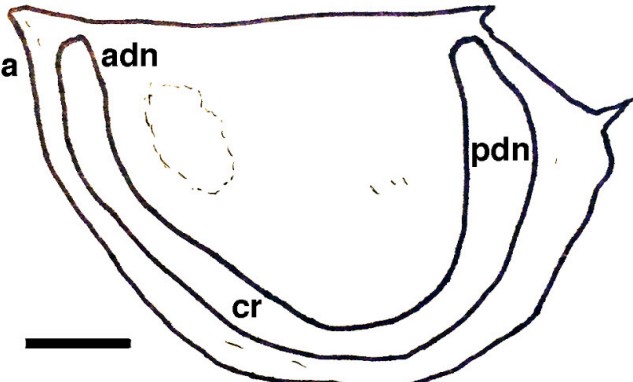

**Figure 25.** *Neokunmingella moroensis*. Anterior to left. Scale bar 200 microns. Abbreviations: **a,** anterodorsal cardinal marginal concavity; **adn**, anterodorsal node; **cr**, connecting ridge; **pdn**, posterodorsal node. Image Credit: Mark McMenamin.

*Nicolarites* Vasilieva, 1994 [99]; Duibianellidae; This genus, known from fragmentary remains of Yakutia (*Nicolarites spasskyi*) and possibly North Greenland [6], consists of hollow tubes, often paired with nodes, that may be the remains of (possibly toxic) bradoriid spines as in *Spinospitella*.

*Oepikaluta* Jones & McKenzie, 1980 [23]; Comptalutidae; Range: late Drumian. Jones and McKenzie [23] considered *Oepikaluta* to be a phosphatocopid, but the genus is considered here to be a bradoriid.

*Onagrocharion* Topper et al. 2007 [100]; Hipponicharionidae; Range: Late Stage 3. The type species of the genus is *Onagrocharion tuberosus.*

Ornamented tube form A; This form from the lower Cambrian of Australia [6] may represent a bradoriid spine. Second-order spines are present.

*Ovaluta* Zhang, 1987 [101]; Bradoriidae; *Ovaluta* has a non-lobate shield [19].

*Paracambria* Huo & Shu, 1985 [53]; Cambriidae; Range: Cambrian.

*Parahoulongdonella* Shu, 1990 [70]; Beyrichonidae; Range: Stage 3, early to middle. *Parahoulongdonella bashanensis* (Figure 26) occurs in the Wirrapowie Limestone in the *M. etheridgei* Zone and the *D. odyssei* Zone, Flinders Ranges, Australia [82]. It is a relatively smooth-surfaced bradoriid lacking nodes and with a relatively wide marginal rim.

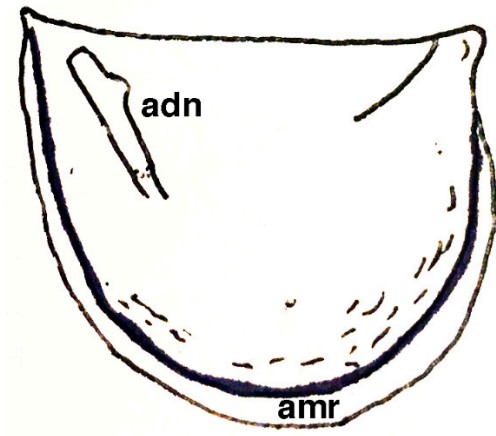

**Figure 26.** *Parahoulongdonella bashanensis*. Anterior to left. Scale bar 200 microns. Abbreviations: **adn**, anterodorsal node; **amr**, admarginal rim. Image Credit: Mark McMenamin.

*Pejonesia* Hinz, 1992 [102]; family unknown [5]; Range: Late Wuliuan. Considered to be a phosphatocopid by Jones and Kruse [58], *Pejonesia* is considered here to be a bradoriid.

*Petrianna* Siveter et al. 1996 [103]; Cambriidae; Range: Late Stage 3. At 17 mm long, *Petrianna* is the largest known bradoriid [2]. *Perianna fulmenata* is a large bradoriid from Greenland that is comparable to certain forms from China [31]. The anastomosing ridges on the shield surface [33,103] are likely reflections of the integumental hemolymph networks [2].

*Phasoia* Hinz-Schallreuter, 1993 [56]; Comptalutidae; Range: Stage 3 to Wuliuan. A sharp continuous ridge forms the lobation in this genus [58]. The surface ornamentation of *Phasoia armini* consists of 10-micron pores with polygonal boundaries [58]. I propose here that the round structures in the interior of most of these polygonal spaces reflect the presence of cell nuclei, hence the reticulate network represents the sheet of cells that was responsible for biomineralization.

*Preaechmina* Shu, 1990 [70]; A spiny bradoriid with a single, central spine that is near the dorsal margin [6].

*Pseudobeyrichona* Cui in Shu, 1990 [70]; Hipponicharionidae; Range: Middle Stage 3. *Pseudobeyrichona monile* (Figure 27) from the Tannenknock Formation in Germany has a relatively smooth, lopsided (postplete) carapace, with distinct denticles ornamenting the marginal rim like a necklace, hence the name after Latin *monile*, necklace [19]. This genus may be synonymous with *Beyrichona* [6]; also assigned to *Neokunmingella* [6]. *Pseudobeyrichona* has also been reported from South China [19].

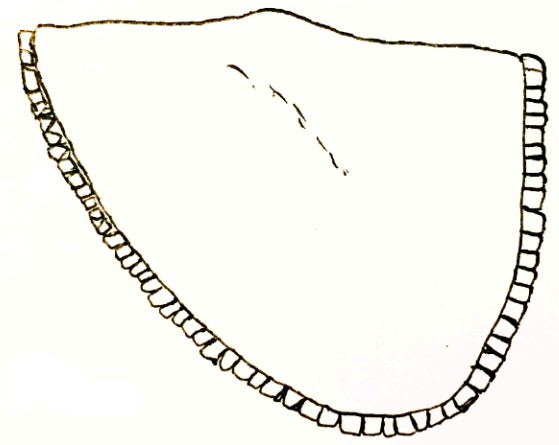

**Figure 27.** *Pseudobeyrichona monile.* Anterior to left. Length of shield 2 mm. Image Credit: Mark McMenamin.

*Quadricona* Topper, Skovsted, Brock & Paterson, 2011 [25]; Hipponicharionidae; *Quadricona* is a bradoriid with a very wide marginal rim and two obtusely pointed, circular nodes on each valve, with the nodes in dorsoanterior and dorsoposterior positions, respectively. Rumply ornamentation occurs on the ventral part of each valve. Conjoined valves viewed in a dorsal view of the shield (along the hinge) show that the nodes project almost perpendicular to the valve surface. The subample *Quadricona madonnae* (Figures 28 and 29) occurs in the Wirrapowie Limestone in the Flinders Ranges, Australia [82]. Each obtusely pointed node in *Quadricona madonnae* resembles the tapered process (one per valve) of the Late Permian myodocopid ostracod *Cypridinelliforma rex* [104]. The anterior and posterior lobes extend further ventrally in the Hipponicharionidae than in the Beyrichonidae [25].

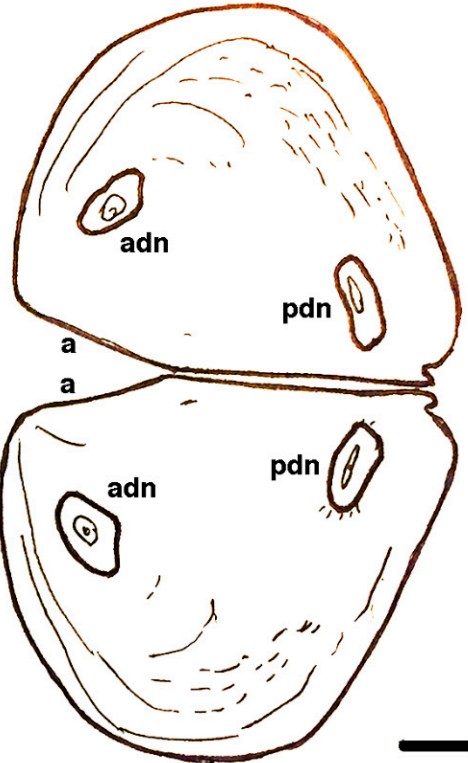

**Figure 28.** *Quadricona madonnae.* Paired shields shown in butterfly position. Anterior to left. Scale bar 200 microns. Abbreviations: **a,** anterodorsal cardinal marginal concavity; **adn**, anterodorsal node; **pdn**, posterodorsal node. Image Credit: Mark McMenamin.

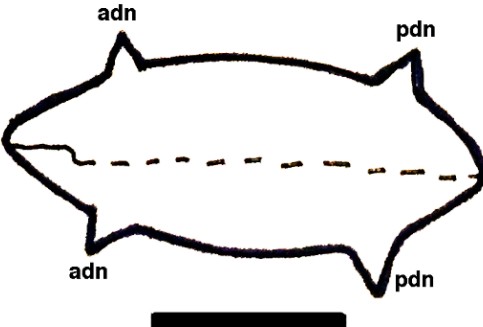

**Figure 29.** *Quadricona madonnae.* Dorsal view of conjoined shields; anterior to left. Scale bar 500 microns. Abbreviations: **adn**, anterodorsal node; **pdn**, posterodorsal node. Image Credit: Mark McMenamin.

*Quetopsis* Hinz-Schallreuter, 1999 [105]; Comptalutidae; Range: Late Wuliuan. This bradoriid has a morphologically simple shield and could be confused with *Houlongdongella*, *Zepaera* and *Flemingopsis* [25].

*Retaluta* Zhang, 2007 [106]; Comptalutidae (Zhexiellidae); Range: Middle Stage 3.

*Salanacus* Grigorieva in Voronin et al. 1982 [107]; *Salanacus* is a spiny possible bradoriid fragment (convex plates with smooth spines) from lower Cambrian strata from Mongolia and China [6].

*Sanlangella* Liu et al. 2008 [108]; Comptalutidae; Range: early Cambrian. *Sanlangella xixiangensis* occurs in the Yanwangbian Formation of southern Shaanxi Province, central China [108]. *Sanlangella* has postplete shields with a large mid-shield node that elongates into a pointed spine. Shield ornament consists of reticulated dimple-like impressions, plus "a single ventrally elongate anterodorsal node" [108].

*Seletellus*; See *Kazakhstanotubulus*.

*Sellula* Wiman, 1902 [109]; Hipponicharionidae; Range: Late Stage 3 to early-late Stage 4 (with question).

*Septadella* Stubblefield, 1933 [110]; Hipponicharionidae; Range: Guzhangian to Ordovician (early Tremadoc). Each valve of *Septadella* has seven elongate lobes [31]. The shields in *Septadella* bear a resemblance to, but are larger than, tetradellid ostracods, also from the Ordovician [5].

*Shangsiella* Li, 1975 [72]; Cambriidae; Range: Late early to Middle part of Stage 3. *Shangsiella* has a connected anterodorsal node, anterodorsal ridge, connecting ridge and posteroventral node, but its posterodorsal node is separate [33]. *Shangsiella* has been considered a synonym of *Chuanbeiella*, *Paracambria* and *Antihipponicharion*.

*Sinskolutella* Melnikova, 1998 [111]; Alutidae; Range: Botomian Stage. In the Siberian Sinsk biota, *Sinskolutella* occurs in the *Bergeroniellus gurarii* Zone [8,112]. *Sinskolutella cuspidata* occurs in the Wirrapowie Limestone in the Flinders Ranges, Australia [28,82]. It has a relatively featureless shield, but does have a short spine on the carapace margin and an inclined aspect similar to that of *Jixinlingella* and *Jiucunella*.

*Spinaluta* Zhang, 2007 [106]; Comptalutidae (Zhexiellidae); Range: Middle part of Stage 3.

*Spinella* Zhang, 2007 [106]; Duibianellidae; Range: Middle part of Stage 3. *Spinella unialata* is considered synonymous with *Mongolitubulus unispinoa* [25].

*Spinospitella* Skovsted, Brock & Paterson 2006 [6]; Duibianellidae; Range: Stages 3 and 4. *Spinospitella coronata* is known from mostly fragmentary remains from the Mernmerna Formation, Bunkers Range, South Australia, Cambrian Stage 4 [6] in a stratigraphic section that also yields the trilobite *Pararaia bunyerooensis* plus five other species of trilobites. Ornamented spines and carapace fragments attributed to *Spinospitella coronata* occur in the Wirrapowie Limestone in the Flinders Ranges, Australia [82]. Complete adult specimens tend to be slightly crumpled, but the juveniles occur intact. The entire outer surface of the valve of *Spinospitella* is covered with fine ornamentation consisting of a reticulate pattern that may associate with tiny nodes. Like *Mongolitubulus*, *Spinospitella* carried ornamented spines, but in addition it has second- and third- order spines that cover the entire bivalve carapace in adult specimens. Second order spines in adults have a "crown of scales" [6] that resemble the scales that cover the large spines of *Mongolitubulus henrikseni*. The resulting primary spines are thus heavily ornamented, giving them a "Christmas Tree" look. As the abundant smaller second-order spines appear to be hollow, I propose here that they represent a defensive array of toxic spines comparable to those of the modern lion fish (*Pterois volitans*) or of caterpillars with toxic spines (such as the hag caterpillar *Phobetron pithecium*) that are used for defense against parasitoid wasps and other predators. *Spinospitella coronata*, with its first order and second order projections, has striking similarity to the Late Permian myodocopid ostracod from Greece *Siveterella flex* [104].

*Sunella* Melnikova et al. 1997 [96]; Range: Late Stage 3 to early Stage 4 (with question). *Sunella* occurs in the east Trans-Baikal region [96].

*Tagra* Hinz-Schallreuter, 1993 [55]; Beyrichonidae; Range: Early Wuliuan. *Tagra koppkai* is placed with question in the Beyrichonidae [19]. The species occurs in the central Anti-Atlas of Morocco (Tagragra Syncline), in the *Ornamentaspis frequens* Biozone [19].

*Tscholponaella* Melnikova in Melnikova et al. 1997 [94]; Range: Late Stage 3 (*parvulus* Biozone). *Tscholponaella orientalis* occurs in the Beshtash Formation of central Kazakhstan [5].

*Tsunyiella* Melnikova in Melnikova et al. 1997 [94]; Svealutidae; Range: Middle Stage 3 to early Stage 4. *Tsunyiella* is a round-oval bradoriid with a wide, smoothly rounded marginal rim. *Tsunyiella gridinae* from the Atdabanian of Kazakhstan shows what are interpreted as integumental hemolymph networks [2].

*Tubuterium* Melnikova, 2000 [96]; Range: Late Stage 3 to middle Guzhangian. A spiny bradoriid. This Middle Cambrian genus is known from Siberia [96], and associated spines resembling *Rushtonites* may be broken-off parts of the *Tubuterium* valves. A reconstruction of *Tubuterium ivantsovi* is notable for portrayal of a bizarre anterodorsal recurved spine [6].

Undetermined svealutid; Svealutidae; An undetermined svealutid occurs in the Franconian Forest of eastern Bavaria, Germany [19].

Unnamed *Indiana*-like bradoriid; Bradoriidae; An unnamed *Indiana*-like bradoriid occurs in the Numayri Member of the Burj Formation, Dead Sea, Jordan (*Kingaspis campbelli* Biozone) [19].

*Usatia* Hinz-Schallreuter, 1993 [56]; family uncertain; Range: Late Wuliuan.

*Ushkarella* Koneva, 1978 [113]; Hippocharionidae; Range: Late Stage 3 to early Stage 4 (with question). *Ushkarella* is from the Botomian of Kazakhstan. Evolutionarily convergent on palaeocopid ostracods, the genus was once placed in the Haoiidae [70] but is now placed in the Hippocharionidae [58]. Possibly a dimorphic taxon [58].

*Uskutchiella* Melnikova in Melnikova et al. 1997 [94]; Range: Late Pabian to early Stage 9. *Uskutchiella sulcata* occurs in the Tandoshka Formation of Gorny Altay [5].

*Vojbokalina* Melnikova in Khazanovich et al. 1984 [114]; Range: Late Drumian to early Guzhangian (with question). Earlier thought to have ostracod affinities [5], this tiny bradoriid is now thought to be convergent with ostracod 'body plan 2' [2].

*Walcottella* Ulrich & Bassler, 1931 [66]; Bradoriidae; Range: Wuliuan; late Stage 3 and Stage 4 (with question) and Drumian to Guzhangian (with question). *Walcottella* is known for its large anteromedian node and punctate valve [7]. Reticulated striae may occur in a concentric pattern around the anterocentral node [68].

*Walcottella*? aff. *apicalis*; Bradoriidae; A *Walcottella*-like bradoriid has been reported from lower Cambrian strata of Sweden [69]. The valve ornamentation is similar to that of the ostracod *Anticythereis* (?) *bopaensis* from the Paleocene of Nigeria [83].

*Wimanicharion* Hinz-Schallreuter, 1993 [56]; Hipponicharionidae; Range: Late Stage 4 to mid-Guzhangian. This genus is known for its "confluent anterior and posterior lobes" [25]. Lobation in *Wimanicharion* develops a single comarginal lobe [31]. *Wimanicharion* aff. *W. matthewi* has been reported from the Valdemiedes Formation (*Ornamentaspis frequens* Biozone) in northern Spain [19].

*Wutingella* Zhang, 1974 [98]; Comptalutidae; Range: Late early to Middle part of Stage 3. *Wutingella binodosa* occurs in the *Abadiella* to *Eoredlichia-Wutingaspis* Biozones of South China [5].

*Yakutingella* Melnikova, 1998 [111]; Kunmingellidae; Range: Botomian Stage. In the Siberian Sinsk biota [8], *Yakutingella* occurs in the *Bergeroniellus gurarii* Zone [112].

*Yaoyingella* Zhang, 1974 [98]; Zhexiellidae; This genus is thought to be a juvenile form of *Comptaluta* [21].

*Yucola* Zhang, 2007 [106]; Range: middle part of early Stage 4.

*Zepaera* Fleming, 1973 [115]; Comptalutidae; Range: Stage 3 to Drumian. *Zepaera* has a wide anterodorsal margin and may develop a finely reticulate surface ornamentation. Anterior lobes form a distinctive omega-shape, but only in juveniles. The omega ridge fades in the adults [25]. The fossils tend to be slightly crumpled, suggesting that the valves were flexible in life. The genus is a "comptalutid bradoriid with distinctive lobes forming W-shaped ridges in the anterior half of juvenile valves. However, in larger growth stages the lobes are effaced . . . and eventually disappear" [6]. The postplete comptalutid *Zepaera jagoi* (Figure 30; Ajax Limestone, Mount Scott Range, Australia) has fine teeth along its margin [25]. Early instars of *Zepaera rete* may be preplete [23,73]. *Zepaera bandeli* occurs in Australia, *T. gibbus* Biozone [5].

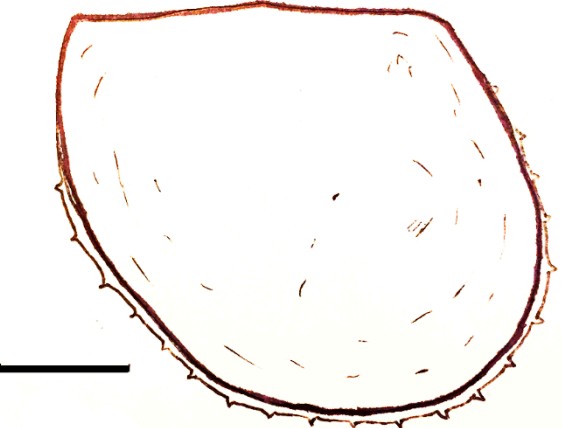

**Figure 30.** *Zepaera jagoi*. Anterior to left. Scale bar 200 microns. Image Credit: Mark McMenamin.

*Zhexiella* Shu, 1990 [70]; Zhexiellidae; Range: Cambrian. A centrodorsal node occurs in *Zhexiella*, and the genus lacks a well-developed rim [6].

## 7. Bradoriid Paleobiogeography

Bradoriids formed a cosmopolitan part of the Cambrian microbenthos, and may have originated in the South China region [5]. Bradoriids have been utilized to define two paleobiogeographic provinces, the European Province and the Four-A's (4A) Province consisting of America, Asia, Australia, and Antarctica [25]. The 4A province has a warmer, more tropical aspect. *Hipponicharion* shows affinities to East and West Gondwana, with occurrences also in Laurentia and South China [31]. *Albrunnicola* is so far restricted to Laurentia, Australia and Antarctica [31]. *Beyrichona* was endemic to Laurentia, Baltica, Avalonia and South China [31]. Facies dependencies and ecological factors, plus an increasingly rich data set such as *Indiana* turning up in Germany, may, however, weaken any biogeographic signals [19]. Nevertheless, the distribution of *Indiana* [19] is in accord with an earlier paleobiogeographic analysis that linked Avalonia, Baltica and Laurentia using the endemic stem brachiopod *Mickwitzia* [116]. *Pseudobeyrichona* from both China and Germany suggests that West Gondwana (Germany) hosted both typically western and eastern bradoriid taxa [19]; thus, this part of Europe has a very long history of being a crossroads between East and West.

The biogeographic significance of *Cambroarchilocus tigris* gen. nov. sp. nov. was first assessed by Siveter and Williams [20], who considered it the southern-most (present day coordinates) occurrence of bradoriids in North America. Streng and coauthors [69] noted that *Cambroarchilocus tigris* gen. nov. sp. nov. "might provide further clues to bradoriid distribution patterns and confirm biogeographic affinities between Baltica and Laurentia." Peel [31] plotted *Cambroarchilocus tigris* gen. nov. sp. nov. (as ?*Albrunnicola*) on a Cambrian Series 2 paleobiogeographic base map, inferring affinities between southern Laurentia and Australia/Antarctica based on the distribution of *Albrunnicola*. Recognition of *Cambroarchilocus* in other parts of the world could potentially generate key paleobiogeographic data for assessing early Cambrian provinciality. I predict here that *Cambroarchilocus* will be discovered in Cambrian strata of Antarctica along with other currently unknown bradoriids of the 4A Province.

## 8. Systematic Paleontology

Repository data: Specimens are located in the IGM repository, Institute of Geology Museum, Departmento de Palaeontología, Cuidad Universitaria, Delegacíon, 04510, México.

Phylum Euarthropoda Lankester, 1904 [117]

Class unknown
Order Bradoriida Raymond, 1935 [1]
Family Hipponicharionidae Sylvester-Bradley, 1961 [7]
Genus *Cambroarchilocus* nov. gen.

**Type species:** monotypic, *Cambroarchilocus tigris* gen. nov. sp. nov.

**Etymology:** Named for the Cambrian System, and to reflect diminutive size as in the hummingbird *Archilocus*, and also for the Greek lyric poet Archilocus (*c.* 680 - *c.* 640 BC), who wrote of a shield that was discarded (*cf.* bradoriid extinction) to be replaced by a new shield (*cf.* appearance of ostracods):

Ἀσπίδι μὲν Σαΐων τις ἀγάλλεται, ἥν παρὰ θάμνῳ
  ἔντος ἀμώμητον κάλλιπον οὐκ ἐθέλων·
αὐτὸν δ' ἔκ μ' ἐσάωσα· τί μοι μέλει ἀσπὶς ἐκείνη;
  Ἐρρέτω· ἐξαῦτις κτήσομαι οὐ κακίω.
One of the Thracians now delights in the shield I discarded
  Unwillingly near a bush, for it was perfectly fine,
But at least I safely escaped. Why should I pine for that shield?
  Let it go. I'll later find another one as good.

[118]

**Diagnosis:** A preplete hipponicharionid bradoriid with a well-delimited posterior lobe with a well-defined sulcus on its anterior side, a weak anterodorsal lobe that overhangs the dorsal margin, and a strongly inflated anterior shield. Appendages unknown.

**Discussion:** *Cambroarchilocus* nov. gen. is most closely allied to three other hipponicharionid bradoriid genera, *Hipponicharion*, *Albrunnicola* and *Beyrichona*. All four genera have the deep triangular valves characteristic for the family (a morphology possibly related to activity of the $Antp^{Tgl}$ allele), but the latter three have a better developed anterodorsal lobe than seen in *Cambroarchilocus*, and none of the three are preplete as is *Cambroarchilocus*.

***Cambroarchilocus tigris* nov. gen. nov. sp.:** Figures 31–37

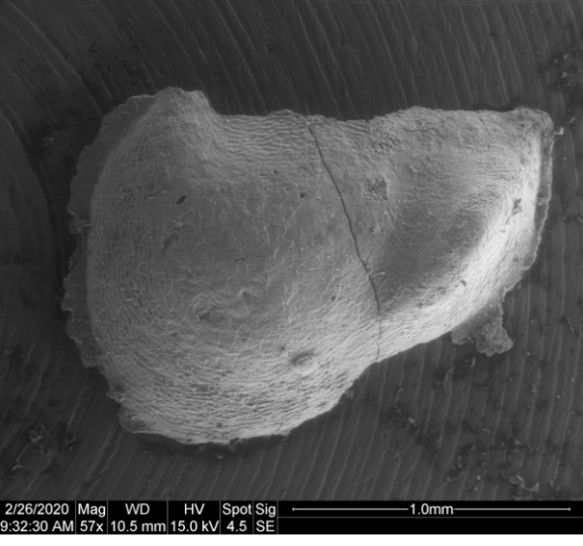

**Figure 31.** *Cambroarchilocus tigris* gen. nov. sp. nov. Holotype. Left shield, anterior to left. IGM 3614 (42). Photo Credit: Mark McMenamin.

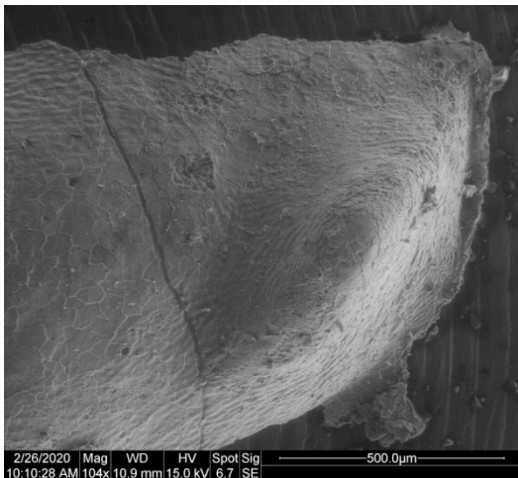

**Figure 32.** *Cambroarchilocus tigris* gen. nov. sp. nov. Holotype. Left shield, detail of posterodorsal node and posterior cardinal corner. Anterior to left. IGM 3614 (42). Photo Credit: Mark McMenamin.

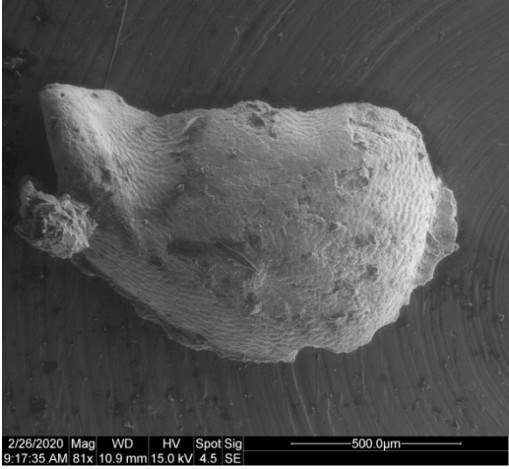

**Figure 33.** *Cambroarchilocus tigris* gen. nov. sp. nov. Paratype. Right shield, anterior to right. Hinge area broken. Irregular lump to at right edge of fossil is an adhering grain of sediment. Arcuate scars along ventral margin attributed here to *Arcuoichnus pierci* nov. ichnogen. nov. ichnosp. IGM 3614 (43). Photo Credit: Mark McMenamin.

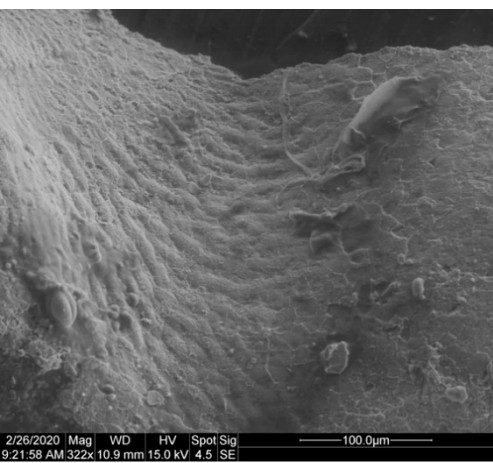

**Figure 34.** *Cambroarchilocus tigris* gen. nov. sp. nov. Paratype. Right shield, anterior to right. Detail of 'fingerprint' ridges in the sulcus anterior to the posterodorsal node. IGM 3614 (43). Photo Credit: Mark McMenamin.

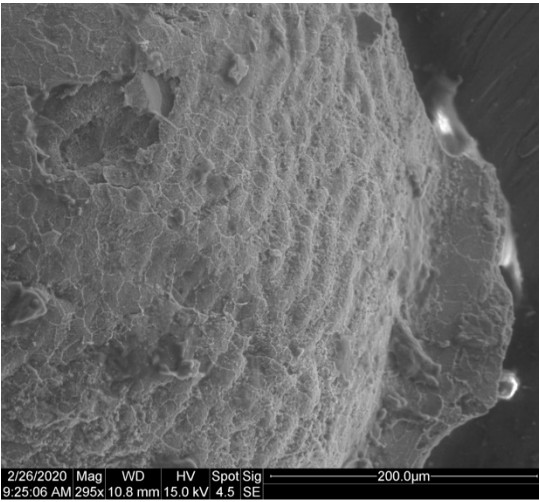

**Figure 35.** *Cambroarchilocus tigris* gen. nov. sp. nov. Paratype. Right shield, anterior to right. Detail of 'fingerprint' ridges and flat marginal rim at the anterior edge of the shield. An example of *Arcuoichnus pierci* nov. ichnogen. nov. ichnosp. occurs at lower right in the image. IGM 3614 (43). Photo Credit: Mark McMenamin.

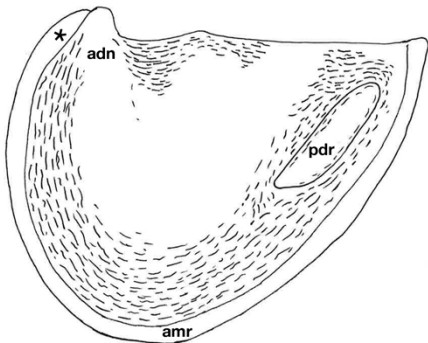

**Figure 36.** *Cambroarchilocus tigris* gen. nov. sp. nov. Reconstruction of left shield, anterior to left. Reconstruction based on both the holotype and paratype. Width of shield 1.5 mm. Abbreviations: **adn**, anterodorsal node; **amr**, admarginal rim; **pdr**, posterodorsal node, * = anterodorsal cardinal marginal concavity. Image Credit: Mark McMenamin.

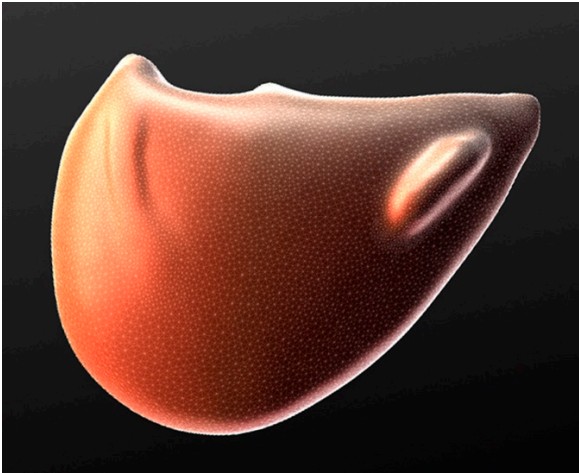

**Figure 37.** *Cambroarchilocus tigris* gen. nov. sp. nov. Reconstruction of left valve, anterior to left (shown here without the admarginal rim). Reconstruction based on both the holotype and paratype. Width of shield 1.5 mm. Image Credit: Mark McMenamin, using Scuptris Alpha 6 by Pixologic, Inc.

**Synonymy:**

| ostracod cf. *Bradoria* | McMenamin and McMenamin 1990 [119] |
|---|---|
| phosphatic ostracod | McMenamin et al. 1994 [120] |
| hipponicharionid | Siveter and Williams 1997 [20] |
| bradoriid ostracod | McMenamin 2001 [121] |
| bradoriid | Zhang 2007 [106] |
| ?*Albrunnicola* | Streng et al. 2008 [69] |
| ?*Albrunnicola* | Peel 2017 [31] |

**Holotype**: Field sample MM-82-49; IGM 3614 (42); left shield.

**Paratype**: Field sample MM-82-49; IGM 3614 (43); right shield.

**Etymology:** From Latin from Greek *tigris* (tiger), for its stripe-like type A1 striation ridge banding, and for the tiger rattlesnake (*Croatus tigris*) that guarded the trail leading into the southern Cerro Rajón field locality in Sonora, México (see p. 201 in reference [122]).

**Diagnosis:** Same as for genus, by monotypy.

**Description**: The *Cambroarchilocus tigris* nov. gen. nov. sp. holotype is a left valve 1.8 mm in greatest dimension. High P and Ca peaks on an SEM-EDS analysis of the holotype, plus the fact that the fossil was recovered by acetic acid maceration of the enclosing archaeocyathan limestone, indicate that the specimens are of hydroxyapatitic composition.

The lateromarginal ridge is relatively wide. Exterior shield ornamentation consists of fine subparallel ridges comparable to the type A1 ('concentric striation') of Siveter and Williams [20].

Although the hinge area is partly damaged, a faint dorsal cusp is visible. The posterior lobe is elongate, and oriented an approximate 50° angle to the vertical posterior edge of the shield. A sulcus occurs just anterior to the posterior lobe.

The paratype is a partially complete right valve 1.4 mm in greatest dimension (Figures 33–35). The anterodorsal lobe is elongate, slightly overhangs the hinge line, and its anterior edge is located adjacent to the broken edge of the valve. There is a medioventral division in the ridge formed by the connection of the ventral slopes of the anterior and posterior lobes. A prominent sulcus occurs posterior to the anterodorsal lobe. As with the holotype, the shield is covered with an ornament of fine, narrowly separated ridges.

Subparallel striations (of type A1) occur as a series of 'fingerprint' ridges in the sulcus depression (Figure 34), and are similar to those appearing on the right valve of a specimen of *Beyrichona papilio* from the Hanford Brook Formation, New Brunswick on Plate 4, Figure 10 of Reference [20].

**Discussion**: The wide lateromarginal ridge in *Cambroarchilocus tigris* nov. gen. nov. sp. suggests that the animal was able to seal its valves with its body inside, and did not require its valves to stay constantly open [28].

In addition, the flat wide marginal rim would also make the shields more effective for flap swimming, with the posterodorsal lobes serving a hydrodynamic function similar to that of the anterior and posterior auricles of the Peruvian scallop *Argopecten purpuratus* [123]. Furthermore, the overhang of the dorsal margin by the anterior lobe in *Cambroarchilocus tigris* gen. nov. sp. nov. would add to the downstroke (assuming the valves could open all the way) of the shield wing, as in the fully opened position the shield wings would have to extend distally to a certain extent to accommodate the overhang of the two valves at the anterior end of the hinge line. This would also tend to stretch the hinge ligament or equivalent to a certain degree, possibly adding power to the downstroke of the hypothesized rowing motion.

The preplete character of *Cambroarchilocus tigris* gen. nov. sp. nov. is confirmed by the presence of an anterodorsal cardinal marginal concavity (Figures 31, 36 and 37). This anterodorsal cardinal marginal concavity is comparable to that of Bengtson's [73] *Hipponicharion* sp. from Kulpara, Parara Limestone, Australia, where the anterodorsal cardinal marginal concavity is clearly visible and marks the anterior end of the shield. A similar anterodorsal cardinal marginal concavity, only weakly concave to straight in this case, is also visible in *Hipponicharion loculatum* (Figure 15). Thus, at least

for the hipponicharionids, recognition of the anterodorsal cardinal marginal concavity is useful for determining the anterior corner of the shield. Indianid bradoriids may also develop this anterodorsal cardinal marginal concavity, as for instance in gen. et sp. indet. G of Bengtson [73].

The shield ornamentation, consisting of type A1 fine ('concentric striation'), resembles that of *Bradoria scrutator* but does not reach the center of the shield [20]. A large crack running diagonally across the holotype shield (Figure 31) was naturally 'healed' by diagenetic recrystallization of hydroxyapatite. Polygonal patterning consisting of thin raised ridges on the surface of both specimens cuts across the A1-type concentric striations and thus appears to be a diagenetic effect although it may have some link to shield microstructure.

**Associated Ichnofossil:** Arcuate perforations along the ventral margin of the paratype (Figures 33 and 35) are of roughly the same size and shape, subtending approximately 40° of arc, and are interpreted here as marks of a microdurophagous predator that attacked a living *Cambroarchilocus tigris* nov. gen. nov. sp. with shields closed for protection. I propose the name *Arcuoichnus pierci* nov. ichnogen. nov. ichnosp. for this trace fossil. This ichnospecies name also applies to minute 'bite marks' on the agnostid *Peronopsis interstricta* [124], as the approximate widths of the respective damage (130 microns for the bradoriid and 200 microns for the agnostid) are comparable. Note that damage by drilling predators has been documented from the ostracod fossil record [83], but not yet from bradoriids.

**Age and Locality Information:** Cambrian Series 2, upper part of Stage 3, lower *Nevadella* Zone, *ca.* 515 million years. This was the point in geologic time when bradoriids reached their maximum diversity (Figure 2).

**Stratigraphic position:** Lower archaeocyathan limestone (MM-82-49) occurring 18.2 m above the base unit 3 of the Puerto Blanco Formation, Cerro Rajón area, Sonora, México [125].

**Collection data:** The fossil was recovered by acetic acid dissolution of limestone collected on March 10–11, 1982 under the auspices of the United States Geological Survey and the Consejo de Recursos Minerales (exercising a federal executive directive [see Plate 1 in reference [122]]). Specimens reside in the Institute of Geology Museum, Departmento de Palaeontología, Cuidad Universitaria, México.

| **Family unknown** |
| :---: |
| Genus *Dielymella* Ulrich and Bassler, 1931 [66] |
| *Dielymella?* sp.: Figure 38 |

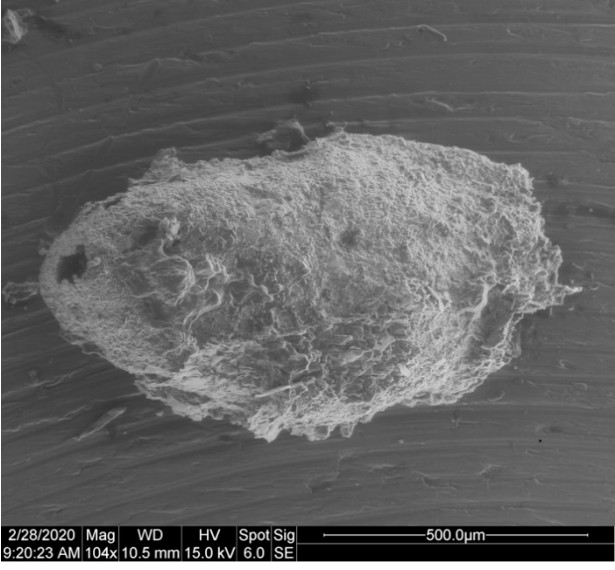

**Figure 38.** *Dielymella?* sp. Phosphatic steinkern of the right shield of an amplete bradoriid with a prominent dorsal cusp. IGM 3614 (44). Photo Credit: Mark McMenamin.

**Material**: Field sample MM-82-49; IGM 3614 (44); steinkern of right shield.

**Description:** This specimen represents the right half of a phosphatic amplete bradoriid steinkern that split in half almost exactly along the plane of commissure. The fossil resembles *Dielymella recticardinalis angustata* in terms of its narrowly-rounded posterior shield, as seen in Plate 9, Figure 5 of reference [20]. It differs from this subspecies by having a more obtuse anterodorsal cardinal angle, by having what appears to be a longer anterodorsal cardinal corner marginal concavity, and by having a more pronounced dorsal cusp than in *Dielymella*, more resembling that of cambriid [33]. Further description must await more complete material.

**Discussion:** Phosphatization of bradoriids is well known [106], but they may also occur as small carbonaceous fossils [126]. The latter mode of preservation in Greenland has preserved first order spines of *Spinospitella*. The fact that bradoriid shields were flexible, plus the known cases of carbonaceous fossil preservation, suggest that their shields contained a significant amount of organic matter, which may have contributed to the formation of phosphatic steinkerns.

**Collection data:** As for previous.

**Age and Locality Information:** As for previous.

**Competing Interests:** Not applicable.

**Acknowledgments:** The author wishes to thank S.M. Awarmik, S. Kiemle, D.L. Schulte McMenamin, J.M. Morales-Ramirez, D. Pierce (hence *Arcuoichnus pierci* nov. ichnogen. nov. ichnosp.), J.H. Stewart, and G. Marchand for assistance with this research.

**Conflicts of Interest:** The authors declare no conflict of interest.

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
