# Peer review of "Bradoriids (Arthropoda) and the Cambrian Diversification"

_geosciences, doi:10.3390/geosciences10040119_

Round 1

Reviewer 1 Report

The paper presents a review of the current state on bradorrids and attempts to synthesize the information thereon as well as presenting some systematic paleontology on a new taxon.

  • The manuscript seems rather convoluted and jumps back and fourth between topics within the paragraphs, seemingly lacking some direction (see marked up PDF).
  • The raw data used to construct the points of changes to studies of bradorrids is not presented so cannot be tested.
  • The writer seems to make some assumptions about the general understanding of the reader regarding the anatomy of bradorrids. I, for example, am not an expert in the group, so have found the new anatomical suggestions, as well as comparisons between families, to be nebulous. I would think a key direction for this manuscript, if it were to be made more useful.
  • The text makes some very bold assumptions about the paleoecology of the group. These suggestions lack much support and this needs some thorough consideration beyond possible functional morphology.
  • The taxonomic consideration is a thorough list of taxa, but lacks any images that would allow the reader to understand more about these animals.
  • I have presented a mark up of the PDF for those areas I felt confident reviewing. I hope these comments are useful.

Unfortunately, at this point, it seems that there is a large amount of work that is needed for the text to be more informative.

Author Response

Reply to Reviewer 1

General comments:

•'lacking some direction'—flow of manuscript improved

•source of raw data in Figures 1, 2 has been made clear

•a shield anatomy diagram for bradoriids is now provided

•'bold assumptions regarding the paleoecology of the group' have been amended

•more illustrations have been added to reinforce taxonomic considerations

Direct comments on the manuscript:

I have accepted (or rendered unnecessary) all changes proposed by Reviewer 1 except the following:

Introduction

•The terms 'waves 1-3' are a good description of the successive waves of bradoriid diversification as proposed here.

Bradoriid Diversification

•Authors of all genera have been provided in the bradoriid genera section. I have elected to not include species author names, as this is not required and would impair the 'readability' of the manuscript.

•It is beyond the scope of this paper to review all Cambrian bivalved groups.

•Figures added in bradoriid genera section

•Bradoriid taxa have not been oversplit in my opinion—as can be seen from the illustrations, the various genera are recognizable and distinct.

Cephalic appendages

•New figure added

Bradoriid Locomotion

•A kinetic reconstuction of bradoriid shield swimming is beyond the scope of this manuscript.

Bradoriid Developmental Biology

•Wording changed here for greater clarity

Bradoriid Genera

•images added

Reviewer 2 Report

A review of Cambrian bradoriids is urgently needed, both due to their importance as the earliest recognizable euarthropod fossils, and due to issues of taxonomic inflation that plague the field. This manuscript is a valuable contribution, however, there are some concerns over the more speculative elements of the paper; suggest reducing some of the speculations to comments or removing them in places (see comments below). Addition of further citations in places would also help strengthen the manuscript, and provide context to some of the interpretations (see comments below).

Line 11; (Abstract); Unnecessary semi-colons in first line, replace with commas.

Line 16; The abstract jumps from a convincing assessment of the very broad-scale importance of bradoriids, to specific information of particular bradoriid morphologies and speculations on their genetic underpinning. Suggest adding at least some transitional text here describing what this study aims to achieve (i.e. a broad review of Cambrian bradoriids), before moving on to some of the highlights.

Line 18; replace semi-colon with comma.

Line 22; ‘Jet-propulsion’ seems an inappropriate phrase. While it may be justified for cephalopods, a clapping of shell valves to propel a swimming arthropod has no analogy with a jet. The phrase appears again at Line 217, Line 221, Line 853, Line 856. Remove the use of this confusing analogy. If an analogy must be used, more appropriate comparisons are various ‘rowing’ motions.

Line 28; supporting references mission. Add suitable references for ‘well-oxygenated shallow-marine sea floor’.

Line 30 – 31; seems an unnecessary statement. All fossils are subject to taphonomic filters. Would perhaps be more useful to simply point out the diverse modes of preservation in which bradoriids are found;

Phosphatised small shelly fossils (SSFs) (e.g., Zhang, X.G., 2007. Phosphatized bradoriids (Arthropoda) from the Cambrian of China. Palaeontographica Abteilung A, pp.93-173.);

Organically-preserved ‘small carbonaceous fossils’ (SCFs) (e.g., Slater, B.J., Willman, S., Budd, G.E. and Peel, J.S., 2018. Widespread preservation of small carbonaceous fossils (SCFs) in the early Cambrian of North Greenland. Geology, 46(2), pp.107-110);

and that they are particularly abundant in certain Lagerstätten deposits such as the Chengjiang (add references to Chengjiang bradoriids).

Line 80 – 87; This paragraph is highly speculative, and in any case is out of place in the Introduction section of this paper. Mixing of analysis, results and interpretations.

General comment; It would be useful to add some more information to the Introduction about other bivalve arthropod groups, comparable to bradoriids (e.g., ostracods, cladocerans, Isoxys-like forms etc.). This would provide background to the discussion raised in Line 102 over the phylogenetic relationships and/or convergence on a bivalve habit among various euarthropods.

Further, it would be useful to provide a range of examples of how different bivalved arthropods can fossilise, in order to give context to the modes of fossilisation of bradoriids. References to cite for different bivalve arthropod groups with a fossil record and their mode of preservation;

Ostracoda;

Xing, L., Sames, B., McKellar, R.C., Xi, D., Bai, M. and Wan, X., 2018. A gigantic marine ostracod (Crustacea: Myodocopa) trapped in mid-Cretaceous Burmese amber. Scientific reports, 8(1), pp.1-9.

(in amber)

Cladocera;

Womack, T., Slater, B.J., Stevens, L.G., Anderson, L.I. and Hilton, J., 2012. First cladoceran fossils from the Carboniferous: palaeoenvironmental and evolutionary implications. Palaeogeography, Palaeoclimatology, Palaeoecology, 344, pp.39-48.

(silicified, in chert)

Bivalved arthropods preserved in the Herefordshire Lagerstätten

(calcitic void infills within carbonate nodules)

Isoxys;

Nielsen, M.L., Rasmussen, J.A. and Harper, D.A., 2017. Sexual dimorphism within the stem-group arthropod Isoxys volucris from the Sirius Passet Lagerstätte, North Greenland. Bulletin of the Geological Society of Denmark., 65, pp.47-58.

(fully-articulated Burgess Shale-type Lagerstätten preservation)

Line 108–113; These hypotheses about bradoriid distribution in relation to Cambrian oxygen levels need some context. Citations needed to show that it is at least feasible to determine oxygen-levels in relation to bradoriid fossil distribution. At the very least, cite some references demonstrating oxygen profiles and distribution on Cambrian shallow-shelves;

Guilbaud, R., Slater, B.J., Poulton, S.W., Harvey, T.H., Brocks, J.J., Nettersheim, B.J. and Butterfield, N.J., 2018. Oxygen minimum zones in the early Cambrian ocean. Geochemical Perspectives Letters, 6, pp.33-38.

Wang, D., Ling, H.F., Struck, U., Zhu, X.K., Zhu, M., He, T., Yang, B., Gamper, A. and Shields, G.A., 2018. Coupling of ocean redox and animal evolution during the Ediacaran-Cambrian transition. Nature communications, 9(1), pp.1-8.

Jin, C., Li, C., Algeo, T.J., Planavsky, N.J., Cui, H., Yang, X., Zhao, Y., Zhang, X. and Xie, S., 2016. A highly redox-heterogeneous ocean in South China during the early Cambrian (∼ 529–514 Ma): Implications for biota-environment co-evolution. Earth and Planetary Science Letters, 441, pp.38-51.

Line 124; change ‘bradoriids’ to ‘Bradoriida’.

Line 184 – 189; Add some discussion here about the level of mineralization of bradoriid valves;

Relevant information included in Vannier et al. 2005 and Betts et al. 2016, which are already included in references.

Line 203 – 217; The discussion here is quite speculative. Suggest condensing this down to a few comments on the proposed ‘flapping’ swimming mode and reduce some of the unnecessary comparisons to aircraft etc.

Line 224; section on bradoriid ‘developmental biology’ is highly speculative, and seems out of place within this paper. An investigation of the potential developmental controls on bradoriid morphology would require an in-depth stand-alone study. Suggest reducing this section to a few comments in the discussion.

Line 647; (Spinospitella); Here it should also be added that organically-preserved carapaces and ornamented first order spines of Spinospitella are known from North Greenland from ‘small carbonaceous fossils’ (SCFs). See Fig. 2 of Slater et al. (2018). Reference; Slater, B.J., Willman, S., Budd, G.E. and Peel, J.S., 2018. Widespread preservation of small carbonaceous fossils (SCFs) in the early Cambrian of North Greenland. Geology, 46(2), pp.107-110.

It would also be worth discussing that this kind of preservation demonstrates that bradoriids clearly had a substantial amount of organic material within their carapaces (relevant for example, to the comments on the flexibility of bradoriid valves in Line 720).

Line 698; The discussion of mimicry here requires substantiation (at minimum a comparison between the outlines of the bradoriid and Mickwitzia brachiopod valves). Is there anything to be gained by mimicking a brachiopod? Where is a citation for the presence of Mickwitzia in the Grammajukku Formation? Could an image and scale comparison of Mickwitzia be provided for comparison?

Line 842; Mention of electrostatic effects seems completely unnecessary. Electrostatic effects are ubiquitous, why raise them here in relation to an extinct arthropod?

Author Response

Replies to Reviewer 2

•line 11  typos fixed

•line 16  transitional text added

•line 18  semicolons removed

•line 22  rowing motion analogy substituted

•line 28  fixed

•lines 30-31  wording changed; reference added later in ms

•lines 80-87  section removed

•General Comment: This information the reviewer provides about the modes of fossilization in other bivalved Cambrian arthropods is interesting but discussion of these cases is beyond the scope of this manuscript.

•lines 108-113  oxygen-levels section removed

•line 124 changed to 'Bradoriida'

•lines 184-189  discussion of degree of biomineralization would be out of place here

•lines 203-217  speculative sentence and aircraft comparison removed

•line 224  discussion revised here and speculative sentence involving Mongolitubulus removed

•line 647  reference added

•line 698  brachiopod mimicry discussion removed

•line 842  'electrostatic' comment removed

Round 2

Reviewer 1 Report

The author has put a considerable amount of effort into improving the text and I commend them for doing so. However, I believe there are more points that need to be addressed for this to be as impactful of a review as possible. These points mostly relate to the figures, as noted in the original review, I am not a bradorrid expert.

  • Figure 1 should have labels on the axes.

  • Figure 2 that appears blurry. This could be due to embedding in the document? This figure would also benefit with being placed against a time calibrated geological time scale such that the reader can appreciate temporally these changes, as at this point it looks as if divisions that are used as of the same width, not indicative of the length of geological divisions. This is also a good place to outline the dominant morphology of each time bin. Give the reader a sense of change through time.

  • One of the major changes that I am pleased to see where taken on board was the addition of figures to make this more useful for the readership. After assessing the drawings, however, I am not sure they are of high enough quality for publication as there are areas there the background looks patchy (see for example figs 6, 14, 16, 24, 25, 26, 27). Is this possibly an artifact of loading the images in place? It would be worth seeing these as a PDF list as opposed to embedding them in the text. Furthermore, these images would be more informative if they were labelled with comments outlining the unique features of the taxa. Finally, associated with these should be images of the holotypes/some form of type, so that this text can be treated as the main reference for bradorrid taxonomy.

I should think if these are addressed the text would be more than informative and indeed one of the seminal reference points for the group.

Author Response

Reply to Reviewer 1 Round 2

•Axes added to Figure 1.

•Figure 2: clarity of image improved; geological time calibration added.

•image patchiness fixed; morphological details indicated on many of the illustrations; .tiff files of all images are available on request

•Images of holotypes (except of course for the holotype of the new taxa) are not required as this paper is primarily focused on the genus level.

•Agreed, this paper may now serve as a seminal reference for the group.

Round 3

Reviewer 1 Report

 I thank the author for considering my changes and having made the appropriate modification to the images.